# HYBRID RANDOM FEATURES

**Krzysztof Choromanski**[*12], **Haoxian Chen**[*†2], **Han Lin**[*†2], **Yuanzhe Ma**[*†2], **Arijit Sehanobish**[*†‡],
**Deepali Jain**[1], **Michael S Ryoo**[1], **Jake Varley**[1], **Andy Zeng**[1], **Valerii Likhosherstov**[13],
**Dmitry Kalashnikov**[1], **Vikas Sindhwani**[1], **Adrian Weller**[34]
[1]Google Brain Robotics, [2]Columbia University, [3]University of Cambridge, [4]The Alan Turing Institute

## ABSTRACT

We propose a new class of random feature methods for linearizing softmax and Gaussian kernels called *hybrid random features* (HRFs) that automatically adapt the quality of kernel estimation to provide most accurate approximation in the defined regions of interest. Special instantiations of HRFs lead to well-known methods such as trigonometric (Rahimi & Recht, 2007) or (recently introduced in the context of linear-attention Transformers) positive random features (Choromanski et al., 2021b). By generalizing Bochner's Theorem for softmax/Gaussian kernels and leveraging random features for compositional kernels, the HRF-mechanism provides strong theoretical guarantees - unbiased approximation and strictly smaller worst-case relative errors than its counterparts. We conduct exhaustive empirical evaluation of HRF ranging from pointwise kernel estimation experiments, through tests on data admitting clustering structure to benchmarking implicit-attention Transformers (also for downstream Robotics applications), demonstrating its quality in a wide spectrum of machine learning problems.

## 1 INTRODUCTION & RELATED WORK

Consider the *softmax* and *Gaussian kernel* functions $\mathrm{K} : \mathbb{R}^{d \times d} \to \mathbb{R}$ defined as follows:

$$\mathrm{SM}(\mathbf{x}, \mathbf{y}) \overset{\text{def}}{=} \exp(\mathbf{x}^\top \mathbf{y}), \qquad \mathrm{K}_{\text{gauss}}(\mathbf{x}, \mathbf{y}) \overset{\text{def}}{=} \exp(-\frac{\|\mathbf{x} - \mathbf{y}\|_2^2}{2}). \tag{1}$$

These two are prominent examples of functions used in the so-called *kernel methods* (Gretton et al., 2005; Zhang et al., 2018) and beyond, i.e. in softmax-sampling (Blanc & Rendle, 2018). Random features (RFs, Rahimi & Recht, 2007; Liu et al., 2020; Peng et al., 2021) yield a powerful mechanism for linearizing and consequently scaling up kernel methods with dot-product kernel decompositions disentangling $\mathbf{x}$ from $\mathbf{y}$ in the formulae for kernel value $\mathrm{K}(\mathbf{x}, \mathbf{y}) \approx \phi(\mathbf{x})^\top \phi(\mathbf{y})$ via data-agnostic probabilistic (random feature) maps $\phi : \mathbb{R}^d \to \mathbb{R}^m$. The tight relationship between softmax and Gaussian kernels given by the transformation $\mathrm{K}_{\text{gauss}}(\mathbf{x}, \mathbf{y}) = \exp(-\frac{\|\mathbf{x}\|_2^2}{2})\mathrm{SM}(\mathbf{x}, \mathbf{y})\exp(-\frac{\|\mathbf{y}\|_2^2}{2})$ provides a mapping of any random feature vector $\phi_{\text{SM}}(\mathbf{u})$ for the softmax kernel to the corresponding one $\phi_{\text{gauss}}(\mathbf{u}) = \exp(-\frac{\|\mathbf{u}\|_2^2}{2})\phi_{\text{SM}}(\mathbf{u})$ for the Gaussian kernel, thus we will focus on the former kernel. The classic random feature map mechanism $\phi_m^{\text{trig}}$ for the softmax kernel, obtained from Bochner's Theorem applied to the Gaussian kernel (Rahimi & Recht, 2007), is of the form:

$$\phi_m^{\text{trig}}(\mathbf{u}) = \frac{1}{\sqrt{m}}\exp(\frac{\|\mathbf{u}\|^2}{2})\left(\sin(\omega_1^\top \mathbf{u}), ..., \sin(\omega_m^\top \mathbf{u}), \cos(\omega_1^\top \mathbf{u}), ..., \cos(\omega_m^\top \mathbf{u})\right)^\top, \tag{2}$$

where $m$ stands for the number of random features and $\omega_1, ..., \omega_m \overset{\text{iid}}{\sim} \mathcal{N}(0, \mathbf{I}_d)$.

The above (common in most downstream use-cases) method for linearizing softmax/Gaussian kernels was recently shown to fail in some of the new impactful applications of scalable kernel methods such as implicit-attention Transformer architectures called Performers (Choromanski et al., 2021b). We denote by MSE the mean squared error of the estimator (i.e. its variance since all estimators considered in this paper are unbiased). The above mechanism struggles to accurately approximate close-to-zero kernel values, as characterized by the particularly large MSE in that region. This is a crucial problem since most of the entries of the attention matrices in Transformers' models are very

---

[*]Equal Contribution, Correspondence to kchoro@google.com.
[†]Authorship in alphabetical order
[‡]Independent researcher. Work done during postdoc at Yale University

small and the approximators need to be particularly accurate there. Otherwise the renormalizers computed to make attention matrices row-stochastic (standard attention normalization procedure in Transformers) would be estimated imprecisely, potentially even by negative values.

The solution to this problem was presented in FAVOR+ mechanism (Choromanski et al., 2021b), where a new positive random feature map for unbiased softmax kernel estimation was applied:

$$\phi_m^{++}(\mathbf{u}) = \frac{1}{\sqrt{2m}} \exp(-\frac{\|\mathbf{u}\|^2}{2}) \left( \exp(\omega_1^\top \mathbf{u}), ..., \exp(\omega_m^\top \mathbf{u}), \exp(-\omega_1^\top \mathbf{u}), ..., \exp(-\omega_m^\top \mathbf{u}) \right)^\top . \quad (3)$$

Even though, very accurate in estimating small softmax kernel values (which turns out to be crucial in making RFs work for Transformers training), this mechanism is characterized by larger MSE for large kernel values. In several applications of softmax kernels (in particular Transformers, where attention matrices typically admit sparse combinatorial structure with relatively few but critical large entries and several close-to-zero ones or softmax sampling) the algorithm needs to process simultaneously very small and large kernel values. The following natural questions arise:

*Is it possible to get the best from both the mechanisms to obtain RFs-based estimators particularly accurate for both very small and large softmax kernel values ? Furthermore, can those estimators be designed to have low variance in more generally pre-defined regions ?*

We give affirmative answers to both of the above questions by constructing a new class of random feature maps techniques called *hybrid random features* or HRFs. Theoretical methods used by us to develop HRFs: (a) provide a unifying perspective, where trigonometric random features from Bochner's Theorem and novel mechanisms proposed in (Choromanski et al., 2021b) are just special corollaries of the more general result from complex analysis, (b) integrate in the original way several other powerful probabilistic techniques such as Goemans-Willimson method (Goemans & Williamson, 2004) and random features for the compositional kernels (Daniely et al., 2017).

We provide detailed theoretical analysis of HRFs, showing in particular that they provide strictly more accurate worst-case softmax kernel estimation than previous algorithms and lead to computational gains. We also conduct their thorough empirical evaluation on tasks ranging from pointwise kernel estimation to downstream problems involving training Transformer-models or even end-to-end robotic controller-stacks including attention-based architectures.

**Related Work:** The literature on different random feature map mechanisms for Gaussian (and thus also softmax) kernel estimation is voluminous. Most focus has been put on reducing the variance of trigonometric random features from (Rahimi & Recht, 2007) via various Quasi Monte Carlo (QMC) methods, where directions and/or lengths of Gaussian vectors used to produce features are correlated, often through geometric conditions such as orthogonality (Choromanski et al., 2017; Rowland et al., 2018; Yu et al., 2016; Choromanski et al., 2019; Choromanski & Sindhwani, 2016). Our HRFs do not compete with those techniques (and can be in fact easily combined with them) since rather than focusing on improving sampling mechanism for a given approximation algorithm, they provide a completely new algorithm. The new application of random features for softmax kernel in Transformers proposed in (Choromanski et al., 2020; 2021b) led to fruitful research on the extensions and limitations of these methods. Schlag et al. (2021) replaced random features by sparse deterministic constructions (no longer approximating softmax kernel). Luo et al. (2021) observed that combining $L_2$-normalization of queries and keys for variance reduction of softmax kernel estimation with FFT-based implementations of relative position encoding and FAVOR+ mechanism from Performers helps in training. Trigonometric random features were applied for softmax sampling in (Rawat et al., 2019). Several other techniques such as Nyström method (Yang et al., 2012; Williams & Seeger, 2000; Rudi et al., 2015) were proposed to construct data-dependent feature representations. Even though, as we show in Sec. 2.4, certain instantiations of the HRF mechanism clearly benefit from some data analysis, our central goal remains an unbiased estimation of the softmax/Gaussian kernels which is no longer the case for those other techniques.

## 2 HYBRID RANDOM FEATURES

### 2.1 PRELIMINARIES

Whenever we do not say explicitly otherwise, all presented lemmas and theorems are new. We start with the following basic definitions and results.

**Definition 2.1** (Kernel with a Random Feature Map Representation). *We say that a kernel function* $\mathrm{K} : \mathbb{R}^d \times \mathbb{R}^d \to \mathbb{R}$ *admits a random feature (RF) map representation if it can be written as*

$$K(\mathbf{x}, \mathbf{y}) = \mathbb{E}_{\omega \sim \Omega} \left[ \sum_{i=1}^{l} \xi_i(\mathbf{x}, \omega) \xi_i(\mathbf{y}, \omega) \right], \tag{4}$$

*for some* $\xi_i : \mathbb{R}^d \times \mathbb{R}^d \to \mathbb{R}$, *and where* $\omega$ *is sampled from some probabilistic distribution* $\Omega \in \mathcal{P}(\mathbb{R}^d)$. *The corresponding random feature map, for a given* $m \in \mathbb{N}$, *is defined as*

$$\phi_m(\mathbf{u}) = \frac{1}{\sqrt{m}} \phi_m^1(\mathbf{u}) \star ... \star \phi_m^l(\mathbf{u}) \in \mathbb{R}^{ml}, \tag{5}$$

*where* $\phi_m^i = (\xi_i(\mathbf{u}, \omega_1), ..., \xi_i(\mathbf{u}, \omega_m))^\top$, $\star$ *stands for vertical concatenation, and* $\omega_1, ..., \omega_m \overset{iid}{\sim} \Omega$.

Random feature maps can be used to unbiasedly approximate corresponding kernels, as follows:

$$\widehat{K}(\mathbf{x}, \mathbf{y}) = \phi_m(\mathbf{x})^\top \phi_m(\mathbf{y}). \tag{6}$$

Using the above notation, a trigonometric random feature $\phi_m^{\text{trig}}$ from Equation 2 can be encoded as applying $l = 2$, $\xi_1(\mathbf{u}, \omega) = \sin(\omega^\top \mathbf{u})$, $\xi_2(\mathbf{u}, \omega) = \cos(\omega^\top \mathbf{u})$. Similarly, positive random features can be encoded as taking $l = 2$ and $\xi_1(\mathbf{u}, \omega) = \frac{1}{\sqrt{2}} \exp(\omega^\top \mathbf{u})$, $\xi_2(\mathbf{u}, \omega) = \frac{1}{\sqrt{2}} \exp(-\omega^\top \mathbf{u})$.

The following result from (Choromanski et al., 2021b) shows that the mean squared error (MSE) of the trigonometric estimator is small for large softmax kernel values and large for small softmax kernel values, whereas an estimator applying positive random features behaves in the opposite way.

Denote by $\widehat{\text{SM}}_m^{\text{trig}}(\mathbf{x}, \mathbf{y})$ an estimator of $\text{SM}(\mathbf{x}, \mathbf{y})$ for $\mathbf{x}, \mathbf{y} \in \mathbb{R}^d$ using trigonometric RFs and $\omega_1, ..., \omega_m \overset{iid}{\sim} \mathcal{N}(0, \mathbf{I}_d)$. Denote by $\widehat{\text{SM}}_m^{++}(\mathbf{x}, \mathbf{y})$ its analogue using positive RFs. We have:

**Lemma 2.2** (positive versus trigonometric RFs). *Take* $\Delta = \mathbf{x} - \mathbf{y}$, $\mathbf{z} = \mathbf{x} + \mathbf{y}$, $f_1(u) = (2m)^{-1} \exp(u^2) \text{SM}^{-2}(\mathbf{x}, \mathbf{y})$, $f_2(u) = (2m)^{-1} \exp(u^2) \text{SM}^2(\mathbf{x}, \mathbf{y})$, $f_3(u) = (1 - \exp(-u^2))^2$. *The MSEs of these estimators are:*

$$\text{MSE}(\widehat{\text{SM}}_m^{\text{trig}}(\mathbf{x}, \mathbf{y})) = f_1(\|\mathbf{z}\|_2) f_3(\|\Delta\|_2), \ \text{MSE}(\widehat{\text{SM}}_m^{++}(\mathbf{x}, \mathbf{y})) = f_2(\|\mathbf{z}\|_2) f_3(\|\mathbf{z}\|_2). \tag{7}$$

## 2.2 THE ALGORITHM

We are ready to present the mechanism of *Hybrid Random Features* (HRFs). Denote by $\mathcal{E} = (\widehat{\text{SM}}^k(\mathbf{x}, \mathbf{y}))_{k=1}^{p+1}$ a list of estimators of $\text{SM}(\mathbf{x}, \mathbf{y})$ (the so-called *base estimators*) and by $\Lambda = (\widehat{\lambda}^k(\mathbf{x}, \mathbf{y}))_{k=1}^{p}$ a list of estimators of $\{\lambda^k(\mathbf{x}, \mathbf{y})\}_{k=1}^{p}$ for some functions $\lambda^k : \mathbb{R}^d \times \mathbb{R}^d \to [0, 1]$, constructed independently from $\mathcal{E}$. Take the following estimator of $\text{SM}(\mathbf{x}, \mathbf{y})$:

$$\widehat{\text{SM}}^{\mathcal{E}, \Lambda}(\mathbf{x}, \mathbf{y}) = \sum_{k=1}^{p} \widehat{\lambda}^k(\mathbf{x}, \mathbf{y}) \widehat{\text{SM}}^k(\mathbf{x}, \mathbf{y}) + \left( 1 - \sum_{k=1}^{p} \widehat{\lambda}^k(\mathbf{x}, \mathbf{y}) \right) \widehat{\text{SM}}^{p+1}(\mathbf{x}, \mathbf{y}) \tag{8}$$

In the next section, we explain in detail how base estimators are chosen. We call $\widehat{\text{SM}}^{\mathcal{E}, \Lambda}(\mathbf{x}, \mathbf{y})$ a hybrid random feature (HRF) estimator of $\text{SM}(\mathbf{x}, \mathbf{y})$ parameterized by $\mathcal{E}, \Lambda$. The role of the $\lambda$-coefficients is to dynamically (based on the input $(\mathbf{x}, \mathbf{y})$) prioritize or deprioritize certain estimators to promote those which are characterized by lower variance for a given input. Note that if elements of $\mathcal{E}$ are unbiased estimators of $\text{SM}(\mathbf{x}, \mathbf{y})$, then trivially $\widehat{\text{SM}}^{\mathcal{E}, \Lambda}(\mathbf{x}, \mathbf{y})$ is also an unbiased estimator of $\text{SM}(\mathbf{x}, \mathbf{y})$. Assume that each $\widehat{\text{SM}}^k(\mathbf{x}, \mathbf{y})$ is of the form $\widehat{\text{SM}}^k(\mathbf{x}, \mathbf{y}) = (\phi_{1,m}^k(\mathbf{x}))^\top \phi_{2,m}^k(\mathbf{y})$ for $\phi_{j,m}^k(\mathbf{u}) = \frac{1}{\sqrt{m}} \phi_{j,m}^{1,k}(\mathbf{u}) \star ... \star \phi_{j,m}^{t_k,k}(\mathbf{u})$, $t_k > 0$ and $\phi_{j,m}^{1,k}, ..., \phi_{j,m}^{t_k,k} : \mathbb{R}^d \to \mathbb{R}^m$, where $j \in \{1, 2\}$. Assume also that $\lambda^k(\mathbf{x}, \mathbf{y})$ can be written as:

$$\lambda^k(\mathbf{x}, \mathbf{y}) = a_k + \mathbb{E}_{\tau \sim \Omega} [\sum_{i=1}^{l_k} f_{1,k}^i(\mathbf{x}, \tau) f_{2,k}^i(\mathbf{y}, \tau)] \tag{9}$$

for some scalars $a_k \in \mathbb{R}$, distribution $\Omega \in \mathcal{P}(\mathbb{R}^d)$ (where $\mathcal{P}(\mathbb{R}^d$ stands for the set of probabilistic distributions on $\mathbb{R}^d$), mappings $\xi_k^i, \eta_k^i : \mathbb{R}^d \times \mathbb{R}^d \to \mathbb{R}$ and that the corresponding estimator $\widehat{\lambda}^k(\mathbf{x}, \mathbf{y}) = \widehat{\lambda}_n^k(\mathbf{x}, \mathbf{y})$ of $\lambda^k(\mathbf{x}, \mathbf{y})$ is of the form:

$$\widehat{\lambda}_n^k(\mathbf{x}, \mathbf{y}) = a_k + (\rho_{1,n}^k(\mathbf{x}))^\top \rho_{2,n}^k(\mathbf{y}) \tag{10}$$

for $\rho_{j,n}^k(\mathbf{u}) = \frac{1}{\sqrt{n}} \rho_{j,n}^{1,k}(\mathbf{u}) \star ... \star \rho_{j,n}^{l_k,k}(\mathbf{u})$, $\rho_{j,n}^{i,k}(\mathbf{u}) = (f_{j,k}^i(\mathbf{u}, \tau_1), ..., f_{j,k}^i(\mathbf{u}, \tau_n))^\top$, $\tau_1, ..., \tau_n \sim \Omega$, and where $j \in \{1, 2\}$. Linearization of the $\lambda$-coefficients given by Equation 9 is crucial to obtain linearization of the hybrid estimators, and consequently random feature map decomposition.

We denote a hybrid estimator using $n$ random features for its base estimators and $m$ to approximate $\lambda$-coefficients as $\widehat{\mathrm{SM}}_{m,n}^{\mathrm{hyb}}$. Furthermore, for two vectors $\mathbf{u}, \mathbf{v}$, we denote their vectorized outer-product by $\mathbf{u} \otimes \mathbf{v}$. Finally, we denote by $\prod^\star$ vectors-concatenation. Estimator $\widehat{\mathrm{SM}}_{m,n}^{\mathrm{hyb}}(\mathbf{x}, \mathbf{y})$ can be rewritten as a dot-product of two (hybrid) random feature vectors, as the next lemma shows.

**Lemma 2.3.** *The HRF estimator $\widehat{\mathrm{SM}}_{m,n}^{\mathrm{hyb}}(\mathbf{x}, \mathbf{y})$ satisfies $\widehat{\mathrm{SM}}_{m,n}^{\mathrm{hyb}}(\mathbf{x}, \mathbf{y}) = \Psi_1(\mathbf{x})^\top \Psi_2(\mathbf{y})$, where $\Psi_j$ for $j \in \{1, 2\}$ is given as $\Psi_j(\mathbf{z}) = \Psi_j^1(\mathbf{z}) \star \Psi_j^2(\mathbf{z}) \star \Psi_j^3(\mathbf{z}) \star \Psi_j^4(\mathbf{z})$ and:*

$$\Psi_j^1(\mathbf{z}) = \prod_{k=1,...,p}^\star \sqrt{\frac{a_k}{m}} \phi_{j,m}^{1,k}(\mathbf{z}) \star ... \star \phi_{j,m}^{t_k,k}(\mathbf{z})$$

$$\Psi_j^2(\mathbf{z}) = \frac{1}{\sqrt{mn}} \prod_{k=1,...,p}^\star \prod_{i,j \in \{1,...,l_k\} \times \{1,...,t_k\}}^\star \rho_{j,n}^{i,k}(\mathbf{z}) \otimes \phi_{j,m}^{j,k}(\mathbf{z})$$

$$\Psi_j^3(\mathbf{z}) = \sqrt{\frac{1 - \sum_{k=1}^p a_k}{m}} \phi_{j,m}^{1,p+1}(\mathbf{z}) \star ... \star \phi_{j,m}^{t_{p+1},p+1}(\mathbf{z})$$

$$\Psi_j^4(\mathbf{z}) = \frac{\mathbf{i}}{\sqrt{mn}} \prod_{k=1,...,p}^\star \prod_{i,j \in \{1,...,l_k\} \times \{1,...,t_{p+1}\}}^\star \rho_{j,n}^{i,k}(\mathbf{z}) \otimes \phi_{j,m}^{j,p+1}(\mathbf{z})$$

(11)

**Bipolar estimators:** A prominent special case of the general hybrid estimator defined above is the one where: $\mathcal{E} = (\widehat{\mathrm{SM}}^{++}(\mathbf{x}, \mathbf{y}), \widehat{\mathrm{SM}}^{\mathrm{trig}}(\mathbf{x}, \mathbf{y}))$. Thus consider the following estimator $\widehat{\mathrm{SM}}_{m,n}^{\mathrm{hyb}}$:

$$\widehat{\mathrm{SM}}_{m,n}^{\mathrm{hyb}}(\mathbf{x}, \mathbf{y}) = \widehat{\lambda}_n(\mathbf{x}, \mathbf{y}) \widehat{\mathrm{SM}}_m^{++}(\mathbf{x}, \mathbf{y}) + (1 - \widehat{\lambda}_n(\mathbf{x}, \mathbf{y})) \widehat{\mathrm{SM}}_m^{\mathrm{trig}}(\mathbf{x}, \mathbf{y}). \tag{12}$$

The question arises whether $\widehat{\mathrm{SM}}_{m,n}^{\mathrm{hyb}}$ defined in such a way can outperform both $\widehat{\mathrm{SM}}_m^{++}$ and $\widehat{\mathrm{SM}}_m^{\mathrm{trig}}$. That of course depends also on the choice of $\lambda : \mathbb{R}^d \times \mathbb{R}^d \to \mathbb{R}$. If we consider a (common) normalized setting where all input vectors have the same norm, we can rewrite $\lambda(\mathbf{x}, \mathbf{y})$ ad $\lambda(\theta_{\mathbf{x},\mathbf{y}}, r)$, where $\theta_{\mathbf{x},\mathbf{y}}$ is an angle between $\mathbf{x}$ and $\mathbf{y}$ and $\|\mathbf{x}\| = \|\mathbf{y}\| = r$. By our previous analysis we know that $\widehat{\mathrm{SM}}_m^{++}$ becomes perfect for $\theta = \pi$ and $\widehat{\mathrm{SM}}_m^{\mathrm{trig}}$ becomes perfect for $\theta = 0$. That suggests particularly simple linear dependence of $\lambda$ on $\theta$ to guarantee vanishing variance for both the critical values: $\theta = 0$ and $\theta = \pi$. It remains to show that such a $\lambda$-coefficient can be linearized. It turns out that this can be done with a particularly simple random feature map mechanism, as we will show later, leading to the so-called angular hybrid variant (see: Section 2.4).

## 2.3 CHOOSING BASE ESTIMATORS FOR HRFs: COMPLEX EXPONENTIAL ESTIMATORS

In this section, we explain how we choose base estimators in Equation 8. We denote by $\mathbf{i}$ a complex number such that $\mathbf{i}^2 = -1$. The following lemma is a gateway to construct unbiased base estimators.

**Lemma 2.4.** *Let $\omega \sim \mathcal{N}(0, \mathbf{I}_d)$. Then for every $\mathbf{z} = (z_1, ..., z_d)^\top \in \mathbb{C}^d$ the following holds:*

$$\mathbb{E}\left[\exp(\omega^\top \mathbf{z})\right] = \exp\left(\frac{\sum_{i=1}^d z_i^2}{2}\right). \tag{13}$$

For a complex vector $\mathbf{z} = (z_1, ..., z_d) \in \mathbb{C}^d$, we denote $\mathbf{z}^2 = \sum_{i=1}^d z_i^2$. Consider $\mathbf{z} = \mathbf{A}\mathbf{x} + (\mathbf{A}^\top)^{-1}\mathbf{y}$ for an invertible (in $\mathbb{C}^{d \times d}$) matrix $\mathbf{A} \in \mathbb{C}^{d \times d}$. Using Lemma 2.4, we get:

$$\exp\left(\frac{(\mathbf{A}\mathbf{x})^2}{2}\right) \exp\left(\frac{((\mathbf{A}^\top)^{-1}\mathbf{y})^2}{2}\right) \mathrm{SM}(\mathbf{x}, \mathbf{y}) = \mathbb{E}[\exp(\omega^\top(\mathbf{A}\mathbf{x} + (\mathbf{A}^\top)^{-1}\mathbf{y}))] \tag{14}$$

Thus for $\Psi_{\mathbf{M}}^m(\mathbf{u}) \overset{\mathrm{def}}{=} \frac{1}{\sqrt{m}} \exp(-\frac{(\mathbf{M}\mathbf{u})^2}{2})(\exp(\omega_1^\top \mathbf{M}\mathbf{u}), ..., \exp(\omega_m^\top \mathbf{M}\mathbf{u}))^\top$ and $\omega_i \sim \mathcal{N}(0, \mathbf{I}_d)$:

$$\mathrm{SM}(\mathbf{x}, \mathbf{y}) = \mathbb{E}[\Psi_{\mathbf{A}}^m(\mathbf{x})^\top \Psi_{(\mathbf{A}^\top)^{-1}}^m(\mathbf{y})]. \tag{15}$$

We obtain the new random feature map mechanism providing unbiased estimation of the softmax kernel. In general it is *asymmetric* since it applies different parameterization for $\mathbf{x}$ and $\mathbf{y}$, i.e. one using $\mathbf{A}$, the other one $(\mathbf{A}^\top)^{-1}$. Asymmetric random features is a simple generalization of the model using the same $\phi$ for both $\mathbf{x}$ and $\mathbf{y}$, presented earlier. The resulting estimators $\widehat{\mathrm{SM}}_m^{\mathrm{cexp}}(\mathbf{x}, \mathbf{y})$, that we call *complex exponential* (CE) will serve as base estimators, and are defined as:

$$\widehat{\mathrm{SM}}_m^{\mathrm{cexp}}(\mathbf{x}, \mathbf{y}) \stackrel{\mathrm{def}}{=} \Psi_\mathbf{A}^m(\mathbf{x})^\top \Psi_{(\mathbf{A}^\top)^{-1}}^m(\mathbf{y}) \tag{16}$$

For $\mathbf{A} = \mathbf{i}\mathbf{I}_d$, CE-estimator becomes trigonometric and for $\mathbf{A} = \mathbf{I}_d$, it becomes $\widehat{\mathrm{SM}}_m^{++}(\mathbf{x}, \mathbf{y})$. Note also that an estimator based on this mechanism has variance equal to zero if

$$\mathbf{x} = -(\mathbf{A}^{-1})(\mathbf{A}^\top)^{-1}\mathbf{y}. \tag{17}$$

In fact we can relax that condition. It is easy to check that for the variance to be zero, we only need:

$$\mathrm{Re}(\mathbf{A})\mathbf{x} + \mathrm{Re}((\mathbf{A}^T)^{-1})\mathbf{y} = 0 \text{ and } \mathrm{Im}(\mathbf{A})\mathbf{x} + \mathrm{Im}((\mathbf{A}^T)^{-1})\mathbf{y} = 0 \tag{18}$$

## 2.4 ADAPTING HRF ESTIMATORS: HOW TO INSTANTIATE HYBRID RANDOM FEATURES

We will use complex exponential estimators from Section 2.3 as base estimators in different insantiations of HRF estimators (that by definition admit random feature map decomposition). The key to the linearization of the HRF estimators is then Equation 9 which implies linearization of the shifted versions of $\lambda$-coefficients (Equation 10) and consequently - desired random feature map decomposition via the mechanism of compositional kernels (details in the Appendix). The questions remains how to construct $\lambda$-coefficients to reduce variance of the estimation in the desired regions of interest.

**Accurate approximation of small and large softmax kernel values simultaneously:** As we explained earlier, in several applications of softmax estimators a desired property is to provide accurate estimation in two antipodal regions - small and large softmax kernel values. This can be done in particular by choosing $p = 1$, $\widehat{\mathrm{SM}}^1 = \widehat{\mathrm{SM}}^{++}$, $\widehat{\mathrm{SM}}^2 = \widehat{\mathrm{SM}}^{\mathrm{trig}}$ and $\lambda(\mathbf{x}, \mathbf{y}) = \frac{\theta_{\mathbf{x},\mathbf{y}}}{\pi}$, where $\theta_{\mathbf{x},\mathbf{y}}$ stands for an angle between $\mathbf{x}$ and $\mathbf{y}$. The key observation is that $\lambda(\mathbf{x}, \mathbf{y})$ can be unbiasedly approximated via the mechanism of sgn random features (following from Goemans-Willimson algorithm (Goemans & Williamson, 2004)) as follows (details in the Appendix, Sec. D):

$$\widehat{\lambda}(\mathbf{x}, \mathbf{y}) = \frac{1}{2} + \rho_n(\mathbf{x})^\top \rho_n(\mathbf{y}), \tag{19}$$

where $\rho_{1,n}(\mathbf{z}) = \rho_{2,n}(\mathbf{z}) = \rho_n(\mathbf{z}) \stackrel{\mathrm{def}}{=} \frac{\mathbf{i}}{\sqrt{2n}}(\mathrm{sgn}(\tau_1^\top \mathbf{z}), ..., \mathrm{sgn}(\tau_n^\top \mathbf{z}))^\top$ for $\tau_1, ..., \tau_n \sim \mathcal{N}(0, \mathbf{I}_d)$ and $\mathbf{i}^2 = -1$. We call the resulting estimator of the softmax kernel *angular hybrid*. As we show in Sec. 3, this estimator is indeed particularly accurate in approximating both small and large softmax kernel values. For instance, for the prominent case when input vectors are of fixed length, its variance is zero for both the smallest ($\theta_{\mathbf{x},\mathbf{y}} = \pi$) and largest ($\theta_{\mathbf{x},\mathbf{y}} = 0$) softmax kernel values (see: Fig. 1).

**Gaussian Lambda Coefficients:** Another tempting option is to instantiate $\lambda$-coefficients with approximate Gaussian kernel values (estimated either via positive or trigonometric random features). In that setting, functions $\widehat{\lambda}^k(\mathbf{x}, \mathbf{y})$ are defined as $\widehat{\lambda}^k(\mathbf{x}, \mathbf{y}) = \exp(-\frac{\|\mathbf{x}+\mathbf{M}_k\mathbf{y}\|_2^2}{2c_k^2})$, for some $\mathbf{M}_1, ..., \mathbf{M}_p \in \mathbb{R}^{d \times d}$ and $c_1, ..., c_p \in \mathbb{R}$. Since this time $\lambda$-coefficients are the values of the Gaussian kernel between vectors $\frac{\mathbf{x}}{c_k}$ and $-\frac{\mathbf{M}_k\mathbf{y}}{c_k}$, we can take $a_k = 0$ and define $\rho_{1,\cdot}^k, \rho_{2,\cdot}^k$ as:

$$\rho_{1,\cdot}^k(\mathbf{x}) = \exp(-\frac{\|\mathbf{x}\|_2^2}{2c_k^2})\phi^{\mathrm{SM}}(\frac{\mathbf{x}}{c_k}) \text{ and } \rho_{2,\cdot}^k(\mathbf{y}) = \exp(-\frac{\|\mathbf{M}_k\mathbf{y}\|_2^2}{2c_k^2})\phi^{\mathrm{SM}}(-\frac{\mathbf{M}_k\mathbf{y}}{c_k}), \tag{20}$$

where $\phi^{\mathrm{SM}}$ is the random feature map corresponding to a particular estimator of the softmax kernel. Note that the resulting hybrid estimator, that we call *Gaussian hybrid* has nice pseudo-combinatorial properties. The coefficients $\lambda^k \in (0, 1]$ fire if $\mathbf{x} \approx -\mathbf{M}_k\mathbf{y}$ (the firing window can be accurately controlled via hyperparameters $c_k$). Thus matrices $\mathbf{M}_k$ can be coupled with the base estimators particularly accurate in the regions, where $\mathbf{x} \approx -\mathbf{M}_k\mathbf{y}$. The mechanism can be thus trivially applied to the setting of accurate approximation of both small and large softmax kernel values for inputs of fixed length, yet it turns out to be characterized by the larger variance than its angular hybrid counterpart and cannot provide variance vanishing property for both $\theta_{\mathbf{x},\mathbf{y}} = 0$ and $\theta_{\mathbf{x},\mathbf{y}} = \pi$.

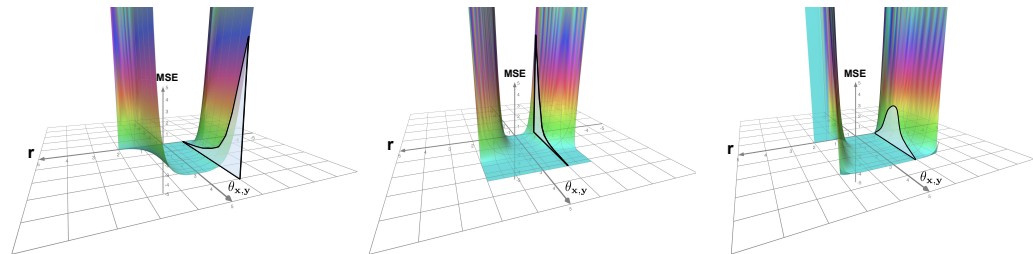

Figure 1: MSEs for three softmax kernel estimators (from left to right): $\widehat{\mathrm{SM}}^{\mathrm{trig}}$, $\widehat{\mathrm{SM}}^{++}$ and angular hybrid for $m = 10, n = 1$ and input of lenghts $r = 5$. MSEs are given as functions of: an angle $\theta_{\mathbf{x},\mathbf{y}} \in [0, \pi]$ between $\mathbf{x}$ and $\mathbf{y}$ and $r$ (symmetrized along $0$ for length axis) . For each plot, we marked in grey its slice for a fixed $r$ to illustrate that only for the angular hybrid estimator, the MSE goes to zero for both $\theta_{\mathbf{x},\mathbf{y}} \to 0$ and $\theta_{\mathbf{x},\mathbf{y}} \to \pi$.

**Adaptation to data admitting clustering structure:** Assume now that the inputs $\mathbf{x}$ come from a distribution $\mathcal{P}(X)$ admitting certain clustering structure with clusters $X_1, ..., X_u \subseteq \mathbb{R}^d$ of centers $\mathbf{x}_1, ..., \mathbf{x}_u \in \mathbb{R}^d$ and that the analogous fact is true for $\mathbf{y}$ with the corresponding clusters $Y_1, ..., Y_w \subseteq \mathbb{R}^d$ and centers $\mathbf{y}_1, ..., \mathbf{y}_w \in \mathbb{R}^d$. For a fixed pair of clusters' centers $(\mathbf{x}_i, \mathbf{y}_j)$, one can associate a complex exponential estimator $\widehat{\mathrm{SM}}^{\mathrm{cexp}}_{(\mathbf{x}_i,\mathbf{y}_j)}$ with the corresponding matrix $\mathbf{A} \in \mathbb{C}^{d \times d}$ (see: Sec. 2.3) that minimizes the following loss, where $|\mathbf{z}|$ for $\mathbf{z} \in \mathbb{C}^d$ is defined as $|\mathbf{z}| = \sqrt{\sum_{i=1}^d |z_i|^2}$:

$$l(\mathbf{A}) = |\mathbf{A}\mathbf{x}_i + (\mathbf{A}^\top)^{-1}\mathbf{y}_j|^2 \tag{21}$$

Note that, by Equation 17, if one can find $\mathbf{A}$ such that $l(\mathbf{A}) = 0$, then $\widehat{\mathrm{SM}}^{\mathrm{cexp}}_{(\mathbf{x}_i,\mathbf{y}_j)}$ makes no error in approximating softmax kernel value on the centers $\mathbf{x}_i, \mathbf{y}_j$. Furthermore, if sampled points are close to the centers and $l(\mathbf{A})$ is small, the corresponding variance of the estimator will also be small. Thus one can think about $\widehat{\mathrm{SM}}^{\mathrm{cexp}}_{(\mathbf{x}_i,\mathbf{y}_j)}$ as an estimator adapted to a pair of clusters $(\mathbf{x}_i, \mathbf{y}_j)$. We can add additional constraints regarding $\mathbf{A}$, i.e. $\mathbf{A} \in \mathbb{R}^{d \times d}$ or $\mathbf{A}$ being diagonal (or both) for closed-form formulae of the optimal $\mathbf{A}$, yet not necessarily zeroing out $l(\mathbf{A})$ (details in the Appendix, Sec. H.3). Complex exponential estimators defined in this way can be combined together, becoming base estimators in the HRF estimator. The corresponding $\lambda$-coefficients can be chosen to be Gaussian as described in the previous paragraph (with optimized matrices $\mathbf{M}_k$ so that a coefficient fires for the corresponding pair of clusters' centers), but there is another much simpler choice of $\lambda$s. We index different base estimators by pairs of clusters' centers $(\mathbf{x}_i, \mathbf{y}_j)$. One can simply take $\rho_1^{(\mathbf{x}_i,\mathbf{y}_j)}(\mathbf{x}) = 1$ if $\mathbf{x}$ belongs to the cluster of center $\mathbf{x}_i$ and $\rho_1^{(\mathbf{x}_i,\mathbf{y}_j)}(\mathbf{x}) = 0$ otherwise. Vectors $\rho_2^{(\mathbf{x}_i,\mathbf{y}_j)}(\mathbf{y})$ (that also become scalars in this scenario) are defined analogously (the $n$ symbol vanishes since $\rho$ is deterministic). Thus effectively the hybrid estimator activates precisely this random feature mechanism that is optimal for the particular pair of clusters. This scheme is feasible if cluster-membership calculation is computationally acceptable, otherwise aforementioned Gaussian coefficients can be applied.

We conclude with an observation that in principle (to reduce the number of base estimators if needed) one can also construct HRF estimators that take base estimators defined in the above way only for the most massive pairs of clusters and add an additional default base estimators (for instance $\widehat{\mathrm{SM}}^{++}$).

## 3 THEORETICAL GUARANTEES

We provide here theoretical guarantees regarding HRF estimators. We focus on the bipolar setting with two base estimators: $\widehat{\mathrm{SM}}^{\mathrm{trig}}$ and $\widehat{\mathrm{SM}}^{++}$. The following is true:

**Theorem 3.1** (MSE of the bipolar hybrid estimator). *Take the bipolar hybrid estimator* $\widehat{\mathrm{SM}}^{\mathrm{hyb}}_{m,n}(\mathbf{x}, \mathbf{y})$, *where* $\widehat{\mathrm{SM}}^{\mathrm{trig}}_m(\mathbf{x}, \mathbf{y})$ *and* $\widehat{\mathrm{SM}}^{++}_m(\mathbf{x}, \mathbf{y})$ *are chosen independently i.e. their random projections are chosen independently (note that we always assume that* $\widehat{\lambda}_n(\mathbf{x}, \mathbf{y})$ *is constructed independently from* $\widehat{\mathrm{SM}}^{\mathrm{trig}}_m(\mathbf{x}, \mathbf{y})$ *and* $\widehat{\mathrm{SM}}^{++}_m(\mathbf{x}, \mathbf{y})$). *Then the following holds:*

$$\mathrm{MSE}(\widehat{\mathrm{SM}}^{\mathrm{hyb}}_{m,n}(\mathbf{x}, \mathbf{y})) = \mathbb{E}[\widehat{\lambda}_n^2(\mathbf{x}, \mathbf{y})]\mathrm{MSE}(\widehat{\mathrm{SM}}^{++}_m(\mathbf{x}, \mathbf{y})) + \mathbb{E}[(1 - \widehat{\lambda}_n(\mathbf{x}, \mathbf{y}))^2]\mathrm{MSE}(\widehat{\mathrm{SM}}^{\mathrm{trig}}_m(\mathbf{x}, \mathbf{y})) \tag{22}$$

*Furthermore, if $\widehat{\mathrm{SM}}_m^{\mathrm{trig}}(\mathbf{x}, \mathbf{y})$ and $\widehat{\mathrm{SM}}_m^{++}(\mathbf{x}, \mathbf{y})$ apply the **exact** same sets of random projections, the mean squared error of the hybrid estimator is further reduced, namely we have:*

$$\mathrm{MSE}(\widehat{\mathrm{SM}}_{m,n}^{\mathrm{hyb}}(\mathbf{x}, \mathbf{y})) = \mathbb{E}[\widehat{\lambda}_n^2(\mathbf{x}, \mathbf{y})]\mathrm{MSE}(\widehat{\mathrm{SM}}_m^{++}(\mathbf{x}, \mathbf{y})) + \mathbb{E}[(1 - \widehat{\lambda}_n(\mathbf{x}, \mathbf{y}))^2]\mathrm{MSE}(\widehat{\mathrm{SM}}_m^{\mathrm{trig}}(\mathbf{x}, \mathbf{y}))$$

$$- \frac{2}{m}\mathrm{SM}^2(\mathbf{x}, \mathbf{y})(1 - \cos(\|\mathbf{x}\|_2^2 - \|\mathbf{y}\|_2^2))\mathbb{E}[\widehat{\lambda}_n(\mathbf{x}, \mathbf{y})(1 - \widehat{\lambda}_n(\mathbf{x}, \mathbf{y}))] \tag{23}$$

*The exact formula on $\mathrm{MSE}(\widehat{\mathrm{SM}}_m^{++}(\mathbf{x}, \mathbf{y}))$ and $\mathrm{MSE}(\widehat{\mathrm{SM}}_m^{\mathrm{trig}}(\mathbf{x}, \mathbf{y}))$ is given in Lemma 2.2.*

We can apply this result directly to the angular hybrid estimator, since:

**Lemma 3.2.** *For the angular hybrid estimator the following holds:*

$$\mathbb{E}[\widehat{\lambda}_n^2(\mathbf{x}, \mathbf{y})] = \frac{\theta_{\mathbf{x},\mathbf{y}}}{\pi}\left(\frac{\theta_{\mathbf{x},\mathbf{y}}}{\pi} - \frac{\theta_{\mathbf{x},\mathbf{y}}}{n\pi} + \frac{1}{n}\right), \mathbb{E}[\widehat{\lambda}_n(\mathbf{x}, \mathbf{y})] = \frac{\theta_{\mathbf{x},\mathbf{y}}}{\pi}. \tag{24}$$

We see that, as mentioned before, the variance of the angular hybrid estimator is zero for both $\theta_{\mathbf{x},\mathbf{y}} = 0$ and $\theta_{\mathbf{x},\mathbf{y}} = \pi$ if inputs $\mathbf{x}, \mathbf{y}$ have the same length. We need one more definition.

**Definition 3.3.** *Assume that the inputs to the estimators are taken from some given bounded set $\mathcal{C} \subseteq \mathbb{R}^d$. For a given estimator $\widehat{\mathrm{SM}}$ on feature vectors $\mathbf{x}, \mathbf{y} \in \mathcal{C}$, we define its max-relative-error with respect to $\mathcal{C}$ as $\epsilon_{\mathcal{C}}(\widehat{\mathrm{SM}}) = \max_{\mathbf{x},\mathbf{y}\in\mathcal{C}} \epsilon_{\mathbf{x},\mathbf{y}}(\widehat{\mathrm{SM}})$, where $\epsilon_{\mathbf{x},\mathbf{y}}(\widehat{\mathrm{SM}}) = \frac{\sqrt{\mathrm{MSE}(\widehat{\mathrm{SM}}(\mathbf{x},\mathbf{y}))}}{\mathrm{SM}(\mathbf{x},\mathbf{y})}$. Denote by $S(r)$ a sphere centered at $0$ and of radius $r$. Define $\epsilon_{\theta,r}(\widehat{\mathrm{SM}}) \stackrel{\mathrm{def}}{=} \epsilon_{\mathbf{x},\mathbf{y}}(\widehat{\mathrm{SM}})$ for $\mathbf{x}, \mathbf{y} \in S(r)$ and such that $\theta = \theta_{\mathbf{x},\mathbf{y}}$ (note that the mean squared errors of the considered estimators depend only on the angle $\theta_{\mathbf{x},\mathbf{y}}$ for $\mathbf{x}, \mathbf{y}$ chosen from a fixed sphere).*

This definition captures the critical observation that in several applications of the softmax kernel estimation, e.g. efficient softmax sampling or linear-attention Transformers (Choromanski et al., 2021b), small relative errors are a much more meaningful measure of the quality of the method than small absolute errors. It also enables us to find hidden symmetries between different estimators:

**Lemma 3.4.** *The following holds:*

$$\epsilon_{\theta,r}(\widehat{\mathrm{SM}}_m^{\mathrm{trig}}) = \epsilon_{\pi-\theta,r}(\widehat{\mathrm{SM}}_m^{++}) = \frac{1}{\sqrt{2m}}\exp(2r^2\sin^2(\frac{\theta}{2}))\left(1 - \exp(-4r^2\sin^2(\frac{\theta}{2}))\right), \tag{25}$$

*and consequently for $W(r) = \exp(2r^2)\left(1 - \exp(-4r^2)\right)$:*

$$\epsilon_{S(r)}(\widehat{\mathrm{SM}}_m^{\mathrm{trig}}) = \epsilon_{S(r)}(\widehat{\mathrm{SM}}_m^{++}) = \lim_{\theta\to\pi}\epsilon_{\theta,r}(\widehat{\mathrm{SM}}_m^{\mathrm{trig}}) = \lim_{\theta\to 0}\epsilon_{\theta,r}(\widehat{\mathrm{SM}}_m^{++}) = \sqrt{\frac{1}{2m}}W(r) \tag{26}$$

Our main result shows that HRF estimators can be applied to reduce the max-relative-error of the previously applied estimators of the softmax kernel. In the next theorem we show that the max-relative-error of the angular hybrid estimator scales as $\frac{1}{r}\exp(2r^2)$ in the length $r$ of its inputs as opposed to $\exp(2r^2)$ as it is the case for $\widehat{\mathrm{SM}}^{\mathrm{trig}}$ and $\widehat{\mathrm{SM}}^{++}$ (see: Lemma 3.4). Furthermore, the max-relative-error scales as $\sqrt{\theta}$ and $\sqrt{\pi - \theta}$ as $\theta \to 0$ and $\theta \to \pi$ respectively, in particular goes to $0$ in both critical cases. This is not true for $\widehat{\mathrm{SM}}^{\mathrm{trig}}$ nor for $\widehat{\mathrm{SM}}^{++}$.

**Theorem 3.5.** *The max-relative-error of the angular hybrid estimator for the inputs $\mathbf{x}, \mathbf{y}$ on the sphere $S(r)$ of radius $r \geq 1$ satisfies for $W(r) = \exp(2r^2)\left(1 - \exp(-4r^2)\right)$:*

$$\epsilon_{S(r)}(\widehat{\mathrm{SM}}_{m,n}^{\mathrm{anghyb}}) \leq \frac{1}{r}\sqrt{\frac{1}{2m}}W(r)\sqrt{\frac{1}{\pi} - \frac{1}{n\pi} + \frac{1}{n\sqrt{\pi}}} \tag{27}$$

*Furthermore, $\lim_{\theta\to 0}\frac{\epsilon_{\theta,r}(\widehat{\mathrm{SM}}_{m,n}^{\mathrm{anghyb}})}{\sqrt{\theta}} = \lim_{\theta\to\pi}\frac{\epsilon_{\theta,r}(\widehat{\mathrm{SM}}_{m,n}^{\mathrm{anghyb}})}{\sqrt{\theta-\pi}} = \sqrt{\frac{1}{2\pi mn}}W(r)$.*

Additional implications of Theorem 3.5, using the fact that $\Theta(mn)$-dimensional HRFs can be constructed in time $O(nd + md + mn)$ (regular RFs need $O(mnd)$), are in the Appendix (Sec. D.4).

## 4 EXPERIMENTS

In this section we conduct exhaustive evaluation of the mechanism of HRFs on several tasks.

## 4.1 Pointwise Softmax Kernel Estimation

We start with ablation studies regarding empirical relative errors of different softmax kernel estimators across different inputs' lengths $r$ and angles $\theta_{\mathbf{x},\mathbf{y}}$. The results are presented in Fig. 2. Ablations over more lengths are presented in the Appendix (Sec. G). The angular hybrid estimator most accurately approximates softmax kernel and has smaller max-relative-error than other estimators.

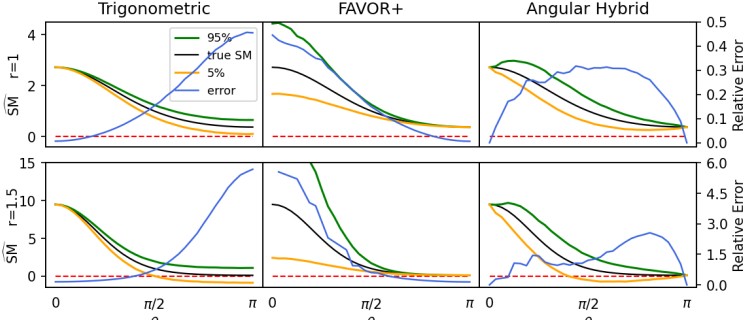

Figure 2: Pointwise estimation of $\mathrm{SM}(\mathbf{x},\mathbf{y})$ for the same-length 64-dim inputs ($r = 1.0$ and $r = 1.5$) and various angles $\theta_{\mathbf{x},\mathbf{y}}$. Red-dotted lines are for marking zero-level. We used $s = 10000$ estimated softmax values in each subplot. The true value and the $5^{th}$ and $95^{th}$ quantile estimated values are shown by the left y-axis, and the empirical relative errors are shown by the right y-axis. Trigonometric estimator and FAVOR+ applied 128 random features. To make fair comparison, for the hybrid variant the configuration leading to the similar number of FLOPS operations per random feature map creation was applied. Similar gains as for the angular are obtained by the Gaussian hybrid variant.

**Comparison with QMC-methods:** Even though, as we explained in Section 1, our algorithm does not compete with various methods for variance reduction based on different sampling techniques (since those methods, as orthogonal to ours, can be easily incorporated into our framework), we decided to compare HRFs also with them. We tested angular hybrid HRF variant as well as well-established QMC baselines: orthogonal random features (ORF)(Yu et al., 2016) (regular variant ORF-reg as well as the one applying Hadamard matrices ORF-Had) and QMCs based on random Halton sequences (Halton-R)(Avron et al., 2016). For HRFs, we tested two sub-variants: with orthogonal (HRF-ort) and regular iid random features (HRF-iid). We computed empirical MSEs by averaging over 100 randomly sampled pairs of vectors from two UCI datasets: wine and Boston. HRFs outperform all other methods, as we see in Table 1.

Table 1: Comparison of different estimators of the softmax kernel on the datapoints from two UCI datasets: wine and Boston in terms of the MSE (measured in $10^{-3}$ units). The non-HRF estimators apply 512 random features and the HRF-ones are set up to match the non-HRFs in terms of the number of FLOPS (for a fair comparison). We also reported standard deviations.

| Datasets | HRF-ort | HRF-iid | ORF-reg | ORF-Had | Halton-R |
|---|---|---|---|---|---|
| Wine | **0.70 ± 0.08** | 0.85 ± 0.05 | 1.00 ± 0.04 | 1.10 ± 0.06 | 4.02 ± 0.1 |
| Boston | **0.72 ± 0.06** | 0.79 ± 0.08 | 1.05 ± 0.03 | 1.14 ± 0.02 | 2.53 ± 0.08 |

## 4.2 Language Modeling

In this section, we apply HRFs to the language modeling task, training a 2-layer LSTM with hidden size $h = 200$ and applying RFs for softmax sampling in training as described in (Rawat et al., 2019). Experimental results are obtained over 10 runs. Experimental details and additional validation results are in the Appendix (Sec. I, Table 4).

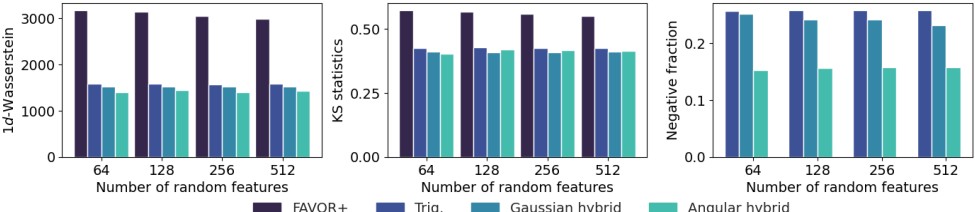

Figure 3: Statistical metrics measuring softmax matrix approximation quality on PennTree Bank. For standard estimators, the number of random features are $64, 128, 256, 512$. To make fair comparison, for the hybrid variants, the configurations leading to the similar number of FLOPS operations per random feature map creation were applied. Negative fractions were not reported for FAVOR+ since by definition they are equal to zero.

**Statistical Metrics.** We compute $1d$-Wasserstein distance and the Kolmogorov-Smirnov (KS) metric (Ramdas et al., 2017) between a true softmax distribution induced by a query on all the classes

and the approximate one given by different RF-mechanisms. We also compute the fraction of all probability estimates with undesired negative values. The results on the PennTree Bank Marcus et al. (1993) are presented in Fig. 3. In the Appendix (Sec. I) we present additional results for the WikiText2 dataset. We see that HRFs lead to most accurate softmax distribution estimation.

### 4.3 TRAINING SPEECH MODELS WITH HRF-CONFORMERS-PERFORMERS

We also tested HRFs on speech models with LibriSpeech ASR corpus (Panayotov et al., 2015). We put implicit Performer's attention into 17-layer Conformer-Transducer encoder (Gulati et al., 2020) and compared softmax kernel estimators in terms of word error rate (WER) metric, commonly used to evaluate speech models. We tested HRFs using angular hybrid variant as well as those clustering-based. For the latter ones, the clusters were created according to the K-means algorithm after first 1000 steps of training (and were frozen afterwards) and corresponding matrices $\mathbf{A}$ were constructed to minimize the loss given in Equation 21. The results are presented in Table 2. HRFs produce smallest WER models and the clustering-based variants fully exercising the most general formulation on HRFs (with general complex estimators) turn out to be the best. Additional details regarding speech experiments as well as additional experiments with clustering-based HRFs are presented in the Appendix (Sec. J and Sec. H.1 respectively).

Table 2: Comparison of WERs of Conformer-Transducer applying different RF-mechanisms for the implicit attention. For methods other than clustering-based HRFs (HRF-C), numbers next to method names define the values of $m$ or $(m, n)$. Method HRF-A stands for the angular hybrid variant. Numbers next to HRF-C correspond to the number of clusters constructed in the query and key space respectively. HRF-C uses 64 random features. We also report standard deviations averaged over 10 different training runs.

|  | HRF-C(3,3) | HRF-C(2, 3) | HRF-C(3,2) | HRF-C(2,2) | HRF-A(16,8) |
|---|---|---|---|---|---|
| WER | $\mathbf{1.72 \pm 0.02}\%$ | $1.75 \pm 0.03\%$ | $1.83 \pm 0.03\%$ | $1.85 \pm 0.04\%$ | $2.03 \pm 0.08\%$ |
|  | HRF-A(8, 8) | FAVOR+ 432 | FAVOR+ 256 | Trig 432 | Trig 256 |
| WER | $2.05 \pm 0.05\%$ | $2.65 \pm 0.06\%$ | $2.77 \pm 0.04\%$ | $3.12 \pm 0.05\%$ | $3.3 \pm 0.06\%$ |

### 4.4 DOWNSTREAM ROBOTICS EXPERIMENTS

In Robotics, we leverage the accuracy of HRFs to obtain inference improvements, critical for on-robot deployment. To abstract from a particular hardware characteristic, we measure it in the number of FLOPS. We conduct two experiments targeting: (a) quadruped locomotion and (b) robotic-arm manipulation. In the former, HRFs are applied as a replacement of the default mechanism using positive random features in the class of implicit-attention architectures for vision processing called *Implicit Attention Policies* (or IAPs) (Choromanski et al., 2021a). In the latter, HRFs become a part of the regular Vision-Performer stack used to process high-resolution (500 x 500 x 3) input. The results are presented in Fig. 4, where HRFs need **3x** fewer FLOPS to obtain same quality policies as baselines. All details regarding both experimental setups are given in the Appendix (Sec. K). Videos of the HRF-trained robotic policies are included in the supplementary material.

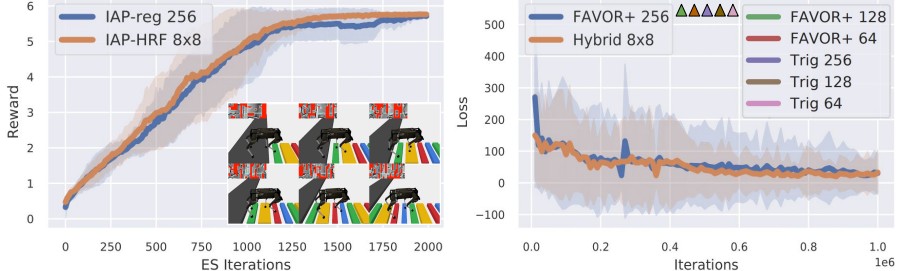

Figure 4: **Left**: Step-stone locomotion task. Comparison of the training curves for: the IAP using angular hybrid estimator of $m = n = 8$ and IAP applying regular FAVOR+ mechanism from (Choromanski et al., 2021b) with $m = 256$. Both final policies are of similar quality, yet HRF-method requires **3x+** fewer FLOPS to run its trained policy. The visualization of the HRF policy in action and its attention (with filtered out pixels marked in red) is in the bottom right corner. **Right:** Similar setting (and conclusions) but for the robotic-arm manipulation task. The additional five regular RF-configurations did not train by producing Nan loss due to large variance of the underlying softmax kernel estimators.

## 5 CONCLUSION

We presented a new class of random feature techniques, called *hybrid random features* for softmax/Gaussian kernel estimation that more accurately approximate these kernels than previous algorithms. We also demonstrated their robustness in a wide range of applications from softmax sampling to Transformers/attention training, also for downstream Robotics tasks.

**Reproducibility Statement:** Section 4 and the Appendix include all experimental details (e.g. hyperparameter setup) needed to reproduce all the results presented in the paper. The part of the code that we could make publicly available can be found in the following github: `https://github.com/HL-hanlin/HRF_ICLR2022`.

**Ethics Statement:** HRFs can be used in principle to train massive Transformer models with large number of parameters and proportional compute resources. Thus they should be used responsibly, in particular given $CO_2$ emission challenges that scientific community tries to address now.

## ACKNOWLEDGEMENTS

AW acknowledges support from a Turing AI Fellowship under grant EP/V025379/1, The Alan Turing Institute, and the Leverhulme Trust via CFI. AS acknowledges AWS compute resources from Onur Kara.

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

## APPENDIX: HYBRID RANDOM FEATURES

## A    PROOF OF LEMMA 2.4

*Proof.* From the independence of $\omega_1, ..., \omega_d$, we get:

$$\mathbb{E}[\exp(\omega^\top \mathbf{z})] = \prod_{i=1}^{d} \mathbb{E}[\exp(\omega_i z_i)] \tag{28}$$

Thus it suffices to show that for $g \sim \mathcal{N}(0, 1)$ and any $z \in \mathbb{C}$ the following holds:

$$\mathbb{E}[\exp(gz)] = \exp(\frac{z^2}{2}) \tag{29}$$

Define $f(z) = \mathbb{E}[\exp(gz)]$. Note that $f(z) = \exp(\frac{z^2}{2})$ for $z = \mathbf{i}x$, where $\mathbf{i}^2 = -1$ and $x \in \mathbb{R}$ which follows from the formula of the characteristic function of the Gaussian distribution. Similarly, $f(z) = \exp(\frac{z^2}{2})$ for $z \in \mathbb{R}$ which follows from the formula of the moment generating function of the Gaussian distribution. We now use the fact from complex analysis that if two analytic functions $\mathbb{C} \to \mathbb{C}$ are identical on uncountably many points then they are equal on $\mathbb{C}$ to complete the proof (we leave to the reader checking that both $f(z)$ and $g(z) \overset{\text{def}}{=} \exp(\frac{z^2}{2})$ are analytic). $\qquad\square$

## B    PROOF OF LEMMA 2.3

*Proof.* The following is true:

$$\widehat{\mathrm{SM}}_{m,n}^{\mathrm{hyb}}(\mathbf{x}, \mathbf{y}) = \sum_{k=1}^{p} \frac{a_k}{m} (\phi_{1,m}^{1,k}(\mathbf{x}) \star ... \star \phi_{1,m}^{t_k,k}(\mathbf{x}))^\top (\phi_{2,m}^{1,k}(\mathbf{y}) \star ... \star \phi_{2,m}^{t_k,k}(\mathbf{y})) +$$

$$\frac{1}{mn} \sum_{k=1}^{p} \sum_{i=1}^{l_k} \sum_{j=1}^{t_k} (\rho_{1,n}^{i,k}(\mathbf{x}) \otimes \phi_{1,m}^{j,k}(\mathbf{x}))^\top (\rho_{2,n}^{i,k}(\mathbf{y}) \otimes \phi_{2,m}^{j,k}(\mathbf{y})) +$$

$$\frac{1 - \sum_{k=1}^{p} a_k}{m} (\phi_{1,m}^{1,p+1}(\mathbf{x}) \star ... \star \phi_{1,m}^{t_{p+1},p+1}(\mathbf{x}))^\top (\phi_{2,m}^{1,p+1}(\mathbf{y}) \star ... \star \phi_{2,m}^{t_{p+1},p+1}(\mathbf{y})) -$$

$$\frac{1}{mn} \sum_{k=1}^{p} \sum_{i=1}^{l_k} \sum_{j=1}^{t_{p+1}} (\rho_{1,n}^{i,k}(\mathbf{x}) \otimes \phi_{1,m}^{j,p+1}(\mathbf{x}))^\top (\rho_{2,n}^{i,k}(\mathbf{y}) \otimes \phi_{2,m}^{j,p+1}(\mathbf{y})). \tag{30}$$

The statement of the lemma follows directly from that. Note that the key trick is an observation that a product of two functions of $\mathbf{x}, \mathbf{y}$ that can be linearized on expectation (i.e. rewritten with $\mathbf{x}$ and $\mathbf{y}$ disentangled) is itself a function that can be linearized on expectation. In the setting where the former are approximated by random features, the latter is approximated by their cartesian product (the random feature mechanism for the compositional kernels from (Daniely et al., 2017)). $\qquad\square$

## C    BIPOLAR HRF ESTIMATORS: PROOF OF THEOREM 3.1

*Proof.* The following holds:

$$\mathrm{Var}(\widehat{\mathrm{SM}}_{m,n}^{\mathrm{hyb}}(\mathbf{x}, \mathbf{y})) = \mathrm{Var}\left(\widehat{\lambda}_n(\theta)\widehat{\mathrm{SM}}_m^{++}(\mathbf{x}, \mathbf{y})\right) + \mathrm{Var}\left((1 - \widehat{\lambda}_n(\theta))\widehat{\mathrm{SM}}_m^{\mathrm{trig}}(\mathbf{x}, \mathbf{y})\right) +$$
$$2\mathrm{Cov}\left(\widehat{\lambda}_n(\theta)\widehat{\mathrm{SM}}_m^{++}(\mathbf{x}, \mathbf{y}), (1 - \widehat{\lambda}_n(\theta))\widehat{\mathrm{SM}}_m^{\mathrm{trig}}(\mathbf{x}, \mathbf{y})\right) \tag{31}$$

We will focus now on the covariance term. To simplify the notation, we will drop index $n$ in $\widehat{\lambda}_n$. Assume first easier to analyze case where $\widehat{\mathrm{SM}}_m^{\mathrm{trig}}(\mathbf{x}, \mathbf{y})$ and $\widehat{\mathrm{SM}}_m^{++}(\mathbf{x}, \mathbf{y})$ are chosen independently. Then the following is true:

$$\text{Cov}\left(\widehat{\lambda}(\theta)\widehat{\text{SM}}_m^{++}(\mathbf{x},\mathbf{y}),(1-\widehat{\lambda}(\theta))\widehat{\text{SM}}_m^{\text{trig}}(\mathbf{x},\mathbf{y})\right) =$$

$$\mathbb{E}[\widehat{\lambda}(\theta)(1-\widehat{\lambda}(\theta))\widehat{\text{SM}}_m^{++}(\mathbf{x},\mathbf{y})\widehat{\text{SM}}_m^{\text{trig}}(\mathbf{x},\mathbf{y})] - \mathbb{E}[\widehat{\lambda}(\theta)\widehat{\text{SM}}_m^{++}(\mathbf{x},\mathbf{y})]\mathbb{E}[(1-\widehat{\lambda}(\theta))\widehat{\text{SM}}_m^{\text{trig}}(\mathbf{x},\mathbf{y})]$$

$$= \left(\mathbb{E}[\widehat{\lambda}(\theta)(1-\widehat{\lambda}(\theta))] - \mathbb{E}[\widehat{\lambda}(\theta)]\mathbb{E}[(1-\widehat{\lambda}(\theta))]\right)\mathbb{E}[\widehat{\text{SM}}_m^{++}(\mathbf{x},\mathbf{y})]\mathbb{E}[\widehat{\text{SM}}_m^{\text{trig}}(\mathbf{x},\mathbf{y})]$$

$$= -(\text{SM}(\mathbf{x},\mathbf{y}))^2\text{Var}(\widehat{\lambda}(\theta)) \tag{32}$$

Now assume that $\widehat{\text{SM}}_m^{\text{trig}}(\mathbf{x},\mathbf{y})$ and $\widehat{\text{SM}}_m^{++}(\mathbf{x},\mathbf{y})$ use the exact same random projections. Then, using similar analysis as before, we get:

$$\text{Cov}\left(\widehat{\lambda}(\theta)\widehat{\text{SM}}_m^{++}(\mathbf{x},\mathbf{y}),(1-\widehat{\lambda}(\theta))\widehat{\text{SM}}_m^{\text{trig}}(\mathbf{x},\mathbf{y})\right) =$$

$$\mathbb{E}[\widehat{\text{SM}}_m^{++}(\mathbf{x},\mathbf{y})\widehat{\text{SM}}_m^{\text{trig}}(\mathbf{x},\mathbf{y})]\mathbb{E}[\widehat{\lambda}(\theta)(1-\widehat{\lambda}(\theta))] - (\text{SM}(\mathbf{x},\mathbf{y}))^2\mathbb{E}[\widehat{\lambda}(\theta)]\mathbb{E}[(1-\widehat{\lambda}(\theta))] \tag{33}$$

This time however it is no longer the case that $\mathbb{E}[\widehat{\text{SM}}_m^{++}(\mathbf{x},\mathbf{y})\widehat{\text{SM}}_m^{\text{trig}}(\mathbf{x},\mathbf{y})]\mathbb{E}[\widehat{\lambda}(\theta)(1-\widehat{\lambda}(\theta))] = \mathbb{E}[\widehat{\text{SM}}_m^{++}(\mathbf{x},\mathbf{y})]\mathbb{E}[\widehat{\text{SM}}_m^{\text{trig}}(\mathbf{x},\mathbf{y})] = (\text{SM}(\mathbf{x},\mathbf{y}))^2$ since $\widehat{\text{SM}}_m^{++}(\mathbf{x},\mathbf{y})$ and $\widehat{\text{SM}}_m^{\text{trig}}(\mathbf{x},\mathbf{y})$ are no longer independent. In order to compute $\mathbb{E}[\widehat{\text{SM}}_m^{++}(\mathbf{x},\mathbf{y})\widehat{\text{SM}}_m^{\text{trig}}(\mathbf{x},\mathbf{y})]\mathbb{E}[\widehat{\lambda}(\theta)(1-\widehat{\lambda}(\theta))]$, we will first introduce useful denotation.

Denote by $\omega_1,...,\omega_m \stackrel{\text{iid}}{\sim} \mathcal{N}(0,\mathbf{I}_d)$ the random projections sampled to construct both $\widehat{\text{SM}}_m^{++}(\mathbf{x},\mathbf{y})$ and $\widehat{\text{SM}}_m^{\text{trig}}(\mathbf{x},\mathbf{y})$. Denote: $Y_i = \cosh(\omega_i^\top(\mathbf{x}+\mathbf{y})) \stackrel{\text{def}}{=} \frac{\exp(\omega_i^\top(\mathbf{x}+\mathbf{y}))+\exp(-\omega_i^\top(\mathbf{x}+\mathbf{y}))}{2}$. We have:

$$\widehat{\text{SM}}_m^{++}(\mathbf{x},\mathbf{y}) = \exp\left(-\frac{\|\mathbf{x}\|^2+\|\mathbf{y}\|^2}{2}\right)\frac{Y_1+...+Y_m}{m} \tag{34}$$

If we denote: $Z_i = \cos(\omega_i^\top(\mathbf{x}-\mathbf{y}))$, then we can write $\widehat{\text{SM}}_m^{\text{trig}}(\mathbf{x},\mathbf{y})$ as:

$$\widehat{\text{SM}}_m^{\text{trig}}(\mathbf{x},\mathbf{y}) = \exp\left(\frac{\|\mathbf{x}\|^2+\|\mathbf{y}\|^2}{2}\right)\frac{Z_1+...+Z_m}{m} \tag{35}$$

We can then rewrite $\mathbb{E}[\widehat{\text{SM}}_m^{++}(\mathbf{x},\mathbf{y})\widehat{\text{SM}}_m^{\text{trig}}(\mathbf{x},\mathbf{y})]$ as:

$$\mathbb{E}[\widehat{\text{SM}}_m^{++}(\mathbf{x},\mathbf{y})\widehat{\text{SM}}_m^{\text{trig}}(\mathbf{x},\mathbf{y})] = \frac{1}{m^2}\left[\sum_{i\neq j}\mathbb{E}[Y_iZ_j] + \sum_{i=1}^m\mathbb{E}[Y_iZ_i]\right] =$$

$$\frac{1}{m^2}\left[\binom{m}{2}(\text{SM}(\mathbf{x},\mathbf{y}))^2 + m\mathbb{E}[\cosh(\omega^\top(\mathbf{x}+\mathbf{y}))\cos(\omega^\top(\mathbf{x}-\mathbf{y}))]\right], \tag{36}$$

where $\omega \sim \mathcal{N}(0,\mathbf{I}_d)$. The equality follows from the unbiasedness of $\widehat{\text{SM}}_m^{\text{trig}}(\mathbf{x},\mathbf{y})$ and $\widehat{\text{SM}}_m^{++}(\mathbf{x},\mathbf{y})$ and the fact that different $\omega_i$ are chosen independently. Thus it remains to compute $\rho = \mathbb{E}[\cosh(\omega^\top(\mathbf{x}+\mathbf{y}))\cos(\omega^\top(\mathbf{x}-\mathbf{y}))]$. Note first that $\rho = \mathbb{E}[\exp(\omega^\top(\mathbf{x}+\mathbf{y}))\cos(\omega^\top(\mathbf{x}-\mathbf{y}))]$ since $-\omega \sim \mathcal{N}(0,\mathbf{I}_d)$ and $\cos$ is an even function. Denote $\mathbf{z} = \mathbf{x}+\mathbf{y}+\mathbf{i}(\mathbf{x}-\mathbf{y})$. We have:

$$\mathbb{E}[\exp(\omega^\top(\mathbf{x}+\mathbf{y}))\cos(\omega^\top(\mathbf{x}-\mathbf{y}))] = \text{Re}\left[\mathbb{E}[\exp(\omega^\top\mathbf{z})]\right] = \text{Re}[\prod_{i=1}^d\exp(\frac{z_i^2}{2})] =$$

$$\text{Re}\left[\exp\left(\frac{\sum_{j=1}^d(x_j+y_j)^2+2\mathbf{i}(x_j^2-y_j^2)-(x_j-y_j)^2}{2}\right)\right] = (\text{SM}(\mathbf{x},\mathbf{y}))^2\cos(\|\mathbf{x}\|_2^2-\|\mathbf{y}\|_2^2) \tag{37}$$

Thus we conclude that:

$$\mathbb{E}[\widehat{\text{SM}}_m^{++}(\mathbf{x},\mathbf{y})\widehat{\text{SM}}_m^{\text{trig}}(\mathbf{x},\mathbf{y})] = (1-\frac{1}{m})(\text{SM}(\mathbf{x},\mathbf{y}))^2 + \frac{1}{m}(\text{SM}(\mathbf{x},\mathbf{y}))^2\cos(\|\mathbf{x}\|_2^2-\|\mathbf{y}\|_2^2) \tag{38}$$

Therefore we get the formulae for the covariance term in both: the setting where random projections of $\widehat{\text{SM}}_m^{\text{trig}}(\mathbf{x}, \mathbf{y})$ and $\widehat{\text{SM}}_m^{++}(\mathbf{x}, \mathbf{y})$ are shared (variant II) and when they are not (variant I). The following is true for $Z = \frac{1}{m}(1 - \cos(\|\mathbf{x}\|_2^2 - \|\mathbf{y}\|_2^2)\mathbb{E}[\widehat{\lambda}(\theta)(1 - \widehat{\lambda}(\theta))]$:

$$\text{Cov}(\widehat{\lambda}(\theta)\widehat{\text{SM}}_m^{++}(\mathbf{x}, \mathbf{y}), (1-\widehat{\lambda}(\theta))\widehat{\text{SM}}_m^{\text{trig}}(\mathbf{x}, \mathbf{y})) = \begin{cases} -(\text{SM}(\mathbf{x}, \mathbf{y}))^2 \text{Var}(\widehat{\lambda}(\theta)) \text{ for variant I} \\ -(\text{SM}(\mathbf{x}, \mathbf{y}))^2(\text{Var}(\widehat{\lambda}(\theta)) + Z) \text{ for variant II} \end{cases} \tag{39}$$

We also have the following:

$$\text{Var}(\widehat{\lambda}(\theta)\widehat{\text{SM}}_m^{++}(\mathbf{x}, \mathbf{y})) = \mathbb{E}[(\widehat{\lambda}(\theta))^2(\widehat{\text{SM}}_m^{++}(\mathbf{x}, \mathbf{y}))^2] - (\mathbb{E}[\widehat{\lambda}(\theta)\widehat{\text{SM}}_m^{++}(\mathbf{x}, \mathbf{y})])^2 =$$

$$\mathbb{E}[(\widehat{\lambda}(\theta))^2]\left(\text{MSE}(\widehat{\text{SM}}_m^{++}(\mathbf{x}, \mathbf{y})) + (\text{SM}(\mathbf{x}, \mathbf{y}))^2\right) - (\mathbb{E}[\widehat{\lambda}(\theta)])^2(\text{SM}(\mathbf{x}, \mathbf{y}))^2 = \tag{40}$$

$$(\text{SM}(\mathbf{x}, \mathbf{y}))^2 \text{Var}(\widehat{\lambda}(\theta)) + \mathbb{E}[(\widehat{\lambda}(\theta))^2]\text{MSE}(\widehat{\text{SM}}_m^{++}(\mathbf{x}, \mathbf{y}))$$

and furthermore (by the analogous analysis):

$$\text{Var}((1-\widehat{\lambda}(\theta))\widehat{\text{SM}}_m^{\text{trig}}(\mathbf{x}, \mathbf{y})) = (\text{SM}(\mathbf{x}, \mathbf{y}))^2\text{Var}(\widehat{\lambda}(\theta)) + \mathbb{E}[(1-\widehat{\lambda}(\theta))^2]\text{MSE}(\widehat{\text{SM}}_m^{\text{trig}}(\mathbf{x}, \mathbf{y})) \tag{41}$$

By putting the derived formula for the above variance terms as well as covariance terms back in the Equation 31, we complete the proof of the theorem (note that the mean squared error of the hybrid estimator is its variance since it is unbiased).

$\square$

# D   ANGULAR HYBRID ESTIMATORS

## D.1   PROOF OF LEMMA 3.2

*Proof.* The formula for the expectation is directly implied by the formula (1) from Sec. 2.1 of Cho & Saul (2012) for the zeroth-order arc-cosine kernel $k_0(\mathbf{x}, \mathbf{y}) \stackrel{\text{def}}{=} 1 - \frac{\theta_{\mathbf{x},\mathbf{y}}}{\pi}$. It also follows directly from Goemans-Williamson algorithm (Goemans & Williamson, 2004). We will now derive the formula for $c = \mathbb{E}[\widehat{\lambda}^2(\mathbf{x}, \mathbf{y})]$. Since $\lambda$ depends only on the angle $\theta = \theta_{\mathbf{x},\mathbf{y}}$, we will refer to it as: $\lambda(\theta)$ and to its estimator as $\widehat{\lambda}(\theta)$. Denote:

$$\phi_n^{\text{ang}}(\mathbf{z}) = \frac{1}{\sqrt{n}}(\text{sgn}(\tau_1^\top \mathbf{z}), ..., \text{sgn}(\tau_n^\top \mathbf{z}))^\top \tag{42}$$

Denote: $X_i = (\phi_n^{\text{ang}}(\mathbf{x}))[i]\phi_n^{\text{ang}}(\mathbf{y})[i]$. We have:

$$\widehat{\lambda}(\theta) = \frac{1}{2}\left(1 - \sum_{i=1}^n X_i\right). \tag{43}$$

Note first that by the construction of $\widehat{\lambda}(\theta)$, we have: $\mathbb{E}[\widehat{\lambda}(\theta)] = \frac{\theta}{\pi}$ and thus: $\mathbb{E}[\sum_{i=1}^n X_i] = 1 - \frac{2\theta}{\pi}$. Therefore we conclude that:

$$c = \frac{1}{4}\mathbb{E}\left[1 - 2\sum_{i=1}^n X_i + \left(\sum_{i=1}^n X_i\right)^2\right] = \frac{1}{4}\left(1 - 2(1 - \frac{2\theta}{\pi}) + \sum_{i=1}^n \mathbb{E}[X_i^2] + \sum_{i \neq j}\mathbb{E}[X_i]\mathbb{E}[X_j]\right)$$

$$= \frac{1}{4}\left(1 - 2(1 - \frac{2\theta}{\pi}) + n \cdot \frac{1}{n^2} + n(n-1) \cdot \frac{1}{n^2}(1 - \frac{2\theta}{\pi})^2\right) = \frac{1}{4}\left(4\frac{\theta}{\pi} + (1 - \frac{1}{n})(1 - \frac{2\theta}{\pi})^2\right)$$

$$= \frac{\theta}{\pi}\left(\frac{\theta}{\pi} - \frac{\theta}{n\pi} + \frac{1}{n}\right) \tag{44}$$

$\square$

## D.2 Explicit formula for the MSE of the angular hybrid estimator

Theorem 3.1 and Lemma 3.2 immediately imply the following result.

**Theorem D.1** (MSE of the angular hybrid estimator). *Take the angular hybrid estimator* $\widehat{\mathrm{SM}}_{m,n}^{\mathrm{anghyb}}(\mathbf{x}, \mathbf{y})$, *where* $\widehat{\mathrm{SM}}_m^{\mathrm{trig}}(\mathbf{x}, \mathbf{y})$ *and* $\widehat{\mathrm{SM}}_m^{++}(\mathbf{x}, \mathbf{y})$ *are chosen independently i.e. their random projections are chosen independently (note that we always assume that* $\widehat{\lambda}(\mathbf{x}, \mathbf{y})$ *is chosen independently from* $\widehat{\mathrm{SM}}_m^{\mathrm{trig}}(\mathbf{x}, \mathbf{y})$ *and* $\widehat{\mathrm{SM}}_m^{++}(\mathbf{x}, \mathbf{y})$). *Then the following holds:*

$$
\mathrm{MSE}(\widehat{\mathrm{SM}}_{m,n}^{\mathrm{anghyb}}(\mathbf{x}, \mathbf{y})) = \frac{\theta}{\pi}\left(\frac{\theta}{\pi} - \frac{\theta}{n\pi} + \frac{1}{n}\right)\frac{1}{2m}\exp(\|\mathbf{z}\|^2)\mathrm{SM}^2(\mathbf{x}, \mathbf{y})(1 - \exp(-\|\mathbf{z}\|^2))^2 +
$$
$$
= \left(1 - \frac{\theta}{\pi}\right)\left(1 - \frac{\theta}{\pi} + \frac{\theta}{n\pi}\right)\frac{1}{2m}\exp(\|\mathbf{z}\|^2)\mathrm{SM}^{-2}(\mathbf{x}, \mathbf{y})(1 - \exp(-\|\Delta\|^2))^2
$$
(45)

*for* $\Delta = \mathbf{x} - \mathbf{y}$ *and* $\mathbf{z} = \mathbf{x} + \mathbf{y}$. *Furthermore, if* $\widehat{\mathrm{SM}}_m^{\mathrm{trig}}(\mathbf{x}, \mathbf{y})$ *and* $\widehat{\mathrm{SM}}_m^{++}(\mathbf{x}, \mathbf{y})$ *apply the **exact** same sets of random projections, the mean squared error of the hybrid estimator is further reduced, namely we have:*

$$
\mathrm{MSE}(\widehat{\mathrm{SM}}_{m,n}^{\mathrm{anghyb}}(\mathbf{x}, \mathbf{y})) = \frac{\theta}{\pi}\left(\frac{\theta}{\pi} - \frac{\theta}{n\pi} + \frac{1}{n}\right)\frac{1}{2m}\exp(\|\mathbf{z}\|^2)\mathrm{SM}^2(\mathbf{x}, \mathbf{y})(1 - \exp(-\|\mathbf{z}\|^2))^2 +
$$
$$
\left(1 - \frac{\theta}{\pi}\right)\left(1 - \frac{\theta}{\pi} + \frac{\theta}{n\pi}\right)\frac{1}{2m}\exp(\|\mathbf{z}\|^2)\mathrm{SM}^{-2}(\mathbf{x}, \mathbf{y})(1 - \exp(-\|\Delta\|^2))^2
$$
$$
- \frac{2}{m}\mathrm{SM}^2(\mathbf{x}, \mathbf{y})(1 - \cos(\|\mathbf{x}\|_2^2 - \|\mathbf{y}\|_2^2))\frac{\theta}{\pi}\left(1 - \frac{1}{n} - \frac{\theta}{\pi} + \frac{\theta}{n\pi}\right)
$$
(46)

Thus if $\|\mathbf{x}\|_2 = \|\mathbf{y}\|_2 = r$ then regardless of whether the same sets of random projections are applied or not, we get:

$$
\mathrm{MSE}(\widehat{\mathrm{SM}}_{m,n}^{\mathrm{anghyb}}(\mathbf{x}, \mathbf{y})) = \frac{\theta}{\pi}\left(\frac{\theta}{\pi} - \frac{\theta}{n\pi} + \frac{1}{n}\right)\frac{1}{2m}\exp(8r^2\cos^2(\frac{\theta}{2}) - 2r^2)\cdot
$$
$$
(1 - \exp(-4r^2\cos^2(\frac{\theta}{2})))^2 + \frac{\theta}{\pi}\left(1 - \frac{1}{n} - \frac{\theta}{\pi} + \frac{\theta}{n\pi}\right)\frac{1}{2m}\exp(2r^2)(1 - \exp(-4r^2\sin^2(\frac{\theta}{2})))^2
$$
(47)

## D.3 Proof of Theorem 3.5

*Proof.* Note first that from the derived above formula of the MSE of the lambda-angular bipolar hybrid estimator and the definition of the max-relative-error, we obtain:

$$
\epsilon_{S(r)}(\widehat{\mathrm{SM}}_{m,n}^{\mathrm{anghyb}}) = \frac{\exp(r^2)}{\sqrt{2m}}\sqrt{\max_{\theta \in [0,\pi]} h_r(\theta)},
$$
(48)

where:

$$
h_r(\theta) = a_r(\theta) + a_r(\pi - \theta)
$$
(49)

and $a_r(\theta)$ is defined as:

$$
a_r(\theta) = \frac{\theta}{\pi}(\frac{\theta}{\pi} - \frac{\theta}{n\pi} + \frac{1}{n})\exp(2r^2\cos(\theta))\left(1 - \exp(-4r^2\cos^2(\frac{\theta}{2}))\right)^2.
$$
(50)

Therefore we have:

$$
\epsilon_{S(r)}(\widehat{\mathrm{SM}}_{m,n}^{\mathrm{anghyb}}) \leq \frac{\exp(r^2)}{\sqrt{m}}\sqrt{\max_{\theta \in [0,\pi]} a_r(\theta)}
$$
(51)

Notice that:

$$
a_r(\theta) \leq b_r(\theta)(1 - \exp(-4r^2))^2
$$
(52)

where:

$$b_r(\theta) = b_r^1(\theta) + b_r^2(\theta) \tag{53}$$

and

$$b_r^1(\theta) = (1 - \frac{1}{n})\frac{\theta^2}{\pi^2}\exp(2r^2\cos(\theta)), \tag{54}$$

$$b_r^2(\theta) = \frac{1}{n} \cdot \frac{\theta}{\pi}\exp(2r^2\cos(\theta)). \tag{55}$$

Therefore:

$$\max_{\theta \in [0,\pi]} b_r(\theta) \le \max_{\theta \in [0,\pi]} b_r^1(\theta) + \max_{\theta \in [0,\pi]} b_r^2(\theta) \tag{56}$$

Denote: $b^1 = \max_{\theta \in [0,\pi]} b_r^1(\theta)$ and $b^2 = \max_{\theta \in [0,\pi]} b_r^2(\theta)$. Note that:

$$\frac{db_r^1(\theta)}{d\theta} = \exp(2r^2\cos(\theta))(1 - \frac{1}{n})\frac{2\theta}{\pi^2}(1 - r^2\theta\sin(\theta)) \tag{57}$$

and

$$\frac{db_r^2(\theta)}{d\theta} = \exp(2r^2\cos(\theta))\frac{1}{n\pi}(1 - 2r^2\theta\sin(\theta)) \tag{58}$$

Thus, from the properties of function: $\theta \to \theta\sin(\theta)$ and the fact that $r \ge 1$, we conclude that both derivatives are first non-negative, then non-positive and then non-negative and that the unique local maximum on the interval $[0,\pi]$ is achieved for $\theta \le \frac{\pi}{2}$. Note also that $b_r^1(\theta), b_r^1(\theta) \ge 0$ and $b_r^1(0) = b_r^2(0) = 0$, $b_r^1(\pi) = (1 - \frac{1}{n})\exp(-2r^2)$, $b_r^2(\pi) = \frac{1}{n}\exp(-2r^2)$. We conclude that global maximum for $b_r^i$ on the interval $[0,\pi]$ for $i = 1, 2$ is achieved either in its unique local maximum on that interval or for $\theta = \pi$. Let us consider first: $b_r^1$. In its local maximum on $[0,\pi]$ we have:

$$\theta^*\sin(\theta^*) = \frac{1}{r^2} \tag{59}$$

Since $\theta \le \sin(\theta) \cdot \frac{\pi}{2}$ on $[0, \frac{\pi}{2}]$, we get:

$$(\theta^*)^2 \le \frac{\pi}{2}\frac{1}{r^2}, \tag{60}$$

i.e.:

$$\theta^* \le \sqrt{\frac{\pi}{2}}\frac{1}{r} \tag{61}$$

Therefore:

$$b_r^1(\theta^*) \le (1 - \frac{1}{n})\frac{1}{2\pi r^2}\exp(2r^2) \ge b_r^1(\pi) \tag{62}$$

We thus conclude that:

$$\max_{\theta \in [0,\pi]} b_r^1(\theta) \le (1 - \frac{1}{n})\frac{1}{2\pi r^2}\exp(2r^2) \tag{63}$$

By the completely analogous analysis applied to $b_r^2$, we obtain:

$$\max_{\theta \in [0,\pi]} b_r^1(\theta) \le \frac{1}{2n\sqrt{\pi}r^2}\exp(2r^2) \tag{64}$$

Now, using Equation 51, Equation 52, and Equation 56, we obtain:

$$\epsilon_{S(r)}(\widehat{\mathrm{SM}})_{m,n}^{\mathrm{anghyb}} \le \frac{\exp(2r^2)}{\sqrt{2m}r}(1 - \exp(-4r^2))\sqrt{\frac{1}{\pi} - \frac{1}{n\pi} + \frac{1}{n\sqrt{\pi}}} \tag{65}$$

and that completes the first part of the proof (proof of Inequality 27). The equations on the limits are directly implied by the fact that:

$$\epsilon_{S(r)}(\widehat{\mathrm{SM}}_{m,n}^{\mathrm{anghyb}}) = \frac{\exp(2r^2)}{\sqrt{2m}}\sqrt{h_r(\theta)}. \tag{66}$$

$\square$

### D.4 Discussion of Theorem 3.5

Theorem 3.5 leads to yet another important conclusion not discussed in the main body of the paper. Note that asymptotically as $\theta \to 0$ or $\theta \to \pi$, the relative error decreases (as a function of $m$ and $n$) as $\frac{1}{\sqrt{mn}} = \Theta(\frac{1}{d_{\text{est}}})$, where $d_{\text{est}}$ stands for the number of random features of the resulting estimator. Note also that for the regular estimator the rate of decrease is also $\Theta(\frac{1}{d_{\text{est}}})$ (here we treat all other arguments of the formula for the relative error as constants; we already know that the dependence on them of the relative error for the HRF estimators is superior to the one for regular estimators). The key difference from the computational point of view is that for the HRF estimator, the random feature map can be constructed in time $O(nd + md + mn)$, whereas for the regular estimator it requires time $O(mnd)$ (see: Sec. L). Thus for the regular estimator random feature map computation is much more expensive. An important application, where random feature map computations takes substantial time of the overall compute time are implicit-attention Transformers, such as Performers (Choromanski et al., 2021b). This is also the setting, where an overwhelming fraction of the approximate softmax kernel values will be extreme (very small or large) since an overwhelming fraction of the attention matrix entries after some training will have extreme values (very small or large). In the setting, where the lengths of queries and keys do not vary much (for instance a prominent case where they all have the same fixed length) that corresponds to angle values $\theta \to \pi$ or $\theta \to 0$. This is exactly the scenario from the second part of Theorem 3.5.

### E Proof of Lemma 3.4

*Proof.* Equation 25 follows directly from Lemma 2.2. Notice that the relative error $\epsilon_{\theta,r}(\widehat{\text{SM}}_m^{\text{trig}})$ is an increasing function of $\sin^2(\frac{\theta}{2})$ and thus is largest for $\theta = \pi$. Plugging in this value into the formula of the relative error gives us the expression from the statement of the lemma. Completely analogous analysis holds for $\widehat{\text{SM}}_m^{++}$. $\square$

### F Gaussian Hybrid Estimators

In this section we will provide more results regarding Gaussian hybrid estimators. We will focus on the bipolar scenario and within this scenario on the so-called *normalized* variant described below.

If $\mathbf{x}$ and $\mathbf{y}$ have fixed $L_2$-norm ($\|\mathbf{x}\|_2 = \|\mathbf{y}\|_2 = r$), we can propose a normalized bipolar Gaussian hybrid estimator such that $\lambda(\mathbf{x}, \mathbf{y}) = 0$ if $\theta_{\mathbf{x},\mathbf{y}} = 0$ and $\lambda(\mathbf{x}, \mathbf{y}) = 1$ if $\theta_{\mathbf{x},\mathbf{y}} = \pi$. Furthermore the variance of that estimator, that we call shortly $\widehat{\text{SM}}_{m,n}^{\text{gausshyb}}$, will be zeroed out for $\theta_{\mathbf{x},\mathbf{y}} = 0$ or $\theta_{\mathbf{x},\mathbf{y}} = \pi$, but not for both.

The coefficient-function $\lambda : \mathbb{R}^d \times \mathbb{R}^d \to \mathbb{R}$ is defined as:

$$\lambda(\mathbf{x}, \mathbf{y}) = \frac{1 - \exp(-\frac{\sigma^2}{2}\|\mathbf{x} - \mathbf{y}\|^2)}{\rho} \tag{67}$$

where $\rho$ is given as:

$$\rho = 1 - \exp(-2\sigma^2 r^2) \tag{68}$$

The hyperparameter $\sigma$ controls the smoothness of the Gaussian kernel.

The exponential in $\lambda$ can be estimated either by trigonometric or positive random features. It is easy to see that for the fixed input lengths, in the former case the variance of the estimator is zero for $\theta_{\mathbf{x},\mathbf{y}} = 0$ and in the latter it is zero for $\theta_{\mathbf{x},\mathbf{y}} = \pi$, but the variance never zeroes out for both $\theta_{\mathbf{x},\mathbf{y}} = 0$ and $\theta_{\mathbf{x},\mathbf{y}} = \pi$. In principle, one can derive entire hierarchy of HRF estimators, where the coefficients in an HRF estimator from one level of hierarchy are estimated with the use of HRF estimators from the higher level. In that context, angular hybrid estimator can be applied to provide approximation of the coefficients of the normalized bipolar Gaussian hybrid estimator leading to variance zeroing out for both $\theta_{\mathbf{x},\mathbf{y}} = 0$ and $\theta_{\mathbf{x},\mathbf{y}} = \pi$. However such nested constructions are not the topic of this paper. From now on we will assume that trigonometric random features are applied to estimate $\lambda$-coefficient, Therefore we have:

$$\widehat{\lambda}(\mathbf{x}, \mathbf{y}) = \frac{1}{\rho} - \frac{1}{n\rho}\sum_{i=1}^{n}\cos(\sigma(\omega_i^\top(\mathbf{x} - \mathbf{y}))) \tag{69}$$

and that leads to the following choice of: $a, \rho_{1,2}, \rho_{2,n}$ from Equation 10:

$$\begin{cases} a = \frac{1}{\rho} \\ \rho_{1,n}(\mathbf{z}) = \rho_{2,n}(\mathbf{z}) = \frac{\mathbf{i}}{\sqrt{n\rho}}(\sin(\tau_1^\top \mathbf{z}), \cos(\tau_1^\top \mathbf{z}), ..., \sin(\tau_n^\top \mathbf{z}), \cos(\tau_n^\top \mathbf{z}))^\top \end{cases} \tag{70}$$

where $\tau_1, ..., \tau_n \sim \mathcal{N}(0, \mathbf{I}_d)$ and $\mathbf{i}^2 = -1$.

Using Theorem 3.1, we conclude that:

**Theorem F.1** (MSE of the normalized bipolar Gaussian estimator). *Take the normalized bipolar Gaussian estimator* $\widehat{\mathrm{SM}}_{m,n}^{\mathrm{gausshyb}}(\mathbf{x}, \mathbf{y})$, *where* $\widehat{\mathrm{SM}}_{m}^{\mathrm{trig}}(\mathbf{x}, \mathbf{y})$ *and* $\widehat{\mathrm{SM}}_{m}^{++}(\mathbf{x}, \mathbf{y})$ *are chosen independently i.e. their random projections are chosen independently (note that we always assume that* $\widehat{\lambda}(\mathbf{x}, \mathbf{y})$ *is chosen independently from* $\widehat{\mathrm{SM}}_{m}^{\mathrm{trig}}(\mathbf{x}, \mathbf{y})$ *and* $\widehat{\mathrm{SM}}_{m}^{++}(\mathbf{x}, \mathbf{y})$). *Denote:* $\Delta = \mathbf{x} - \mathbf{y}$. *Then the following holds:*

$$\mathrm{MSE}(\widehat{\mathrm{SM}}_{m,n}^{\mathrm{gausshyb}}(\mathbf{x}, \mathbf{y})) = \mathbb{E}[\widehat{\lambda}^2(\mathbf{x}, \mathbf{y})]\mathrm{MSE}(\widehat{\mathrm{SM}}_{m}^{++}(\mathbf{x}, \mathbf{y})) + \mathbb{E}[(1 - \widehat{\lambda}(\mathbf{x}, \mathbf{y}))^2]\mathrm{MSE}(\widehat{\mathrm{SM}}_{m}^{\mathrm{trig}}(\mathbf{x}, \mathbf{y})) \tag{71}$$

*where*

$$\mathbb{E}[(\widehat{\lambda}(\mathbf{x}, \mathbf{y}))^2] = \frac{1}{\rho^2}(1 - \exp(-\frac{\sigma^2}{2}\|\Delta\|^2))^2 + \frac{1}{2\rho^2 n}(1 - \exp(-\sigma^2\|\Delta\|^2))^2 \tag{72}$$

$$\mathbb{E}[(1 - \widehat{\lambda}(\mathbf{x}, \mathbf{y}))^2] = \frac{1}{\rho^2}((1 - \rho) - \exp(-\frac{\sigma^2}{2}\|\Delta\|^2))^2 + \frac{1}{2\rho^2 n}(1 - \exp(-\sigma^2\|\Delta\|^2))^2. \tag{73}$$

*Furthermore, if* $\widehat{\mathrm{SM}}_{m}^{\mathrm{trig}}(\mathbf{x}, \mathbf{y})$ *and* $\widehat{\mathrm{SM}}_{m}^{++}(\mathbf{x}, \mathbf{y})$ *apply the **exact** same sets of random projections, the mean squared error of the hybrid estimator remains the same.*

*Proof.* We can calculate $\mathbb{E}[(\widehat{\lambda}(\mathbf{x}, \mathbf{y}))^2]$ and $\mathbb{E}[(1 - \widehat{\lambda}(\mathbf{x}, \mathbf{y}))^2]$ as follows:

$$\mathbb{E}[(1 - \widehat{\lambda}(\mathbf{x}, \mathbf{y}))^2] = \frac{1}{\rho^2}\mathbb{E}[(\rho - 1 + \frac{1}{n}\sum_{i=1}^{n}\cos(\sigma\omega^\top(\mathbf{x} - \mathbf{y})))^2]$$

$$= \frac{(1 - \rho)^2}{\rho^2} + \frac{2(\rho - 1)}{\rho^2}\mathbb{E}[\cos(\sigma\omega^\top(\mathbf{x} - \mathbf{y}))] + \frac{1}{\rho^2}\mathbb{E}[(\frac{1}{k}\sum_{i=1}^{n}\cos(\sigma\omega^\top(\mathbf{x} - \mathbf{y})))^2]$$

$$= \frac{(1 - \rho)^2}{\rho^2} + \frac{2(\rho - 1)}{\rho^2}\exp(-\frac{\sigma^2}{2}\|\Delta\|^2) + \frac{1}{\rho^2}\exp(-\sigma^2\|\Delta\|^2) + \frac{1}{2\rho^2 n}(1 - \exp(-\sigma^2\|\Delta\|^2))^2$$

$$= \frac{1}{\rho^2}((1 - \rho) - \exp(-\frac{\sigma^2}{2}\|\Delta\|^2))^2 + \frac{1}{2\rho^2 n}(1 - \exp(-\sigma^2\|\Delta\|^2))^2 \tag{74}$$

$$\mathbb{E}[(\widehat{\lambda}(\mathbf{x}, \mathbf{y}))^2] = 1 - 2\mathbb{E}[1 - \widehat{\lambda}(\mathbf{x}, \mathbf{y}] + \mathbb{E}[(1 - \widehat{\lambda}(\mathbf{x}, \mathbf{y}))^2]$$

$$= -1 + \frac{2}{\rho} - \frac{2}{\rho}\exp(-\frac{\sigma^2}{2}\|\Delta\|^2) + \mathbb{E}[(1 - \widehat{\lambda}(\mathbf{x}, \mathbf{y}))^2] \tag{75}$$

$$= \frac{1}{\rho^2}(1 - \exp(-\frac{\sigma^2}{2}\|\Delta\|^2))^2 + \frac{1}{2\rho^2 n}(1 - \exp(-\sigma^2\|\Delta\|^2))^2$$

By plugging in these two formulae in the expression for MSE, we obtain the first half of the theorem.

Note that if $\widehat{\text{SM}}_m^{\text{trig}}(\theta, r)$ and $\widehat{\text{SM}}_m^{++}(\theta, r)$ apply the exact same sets of random projections, then $(1 - \cos(\|\mathbf{x}\|_2^2 - \|\mathbf{y}\|_2^2))$ in $\text{MSE}(\widehat{\text{SM}}_{m,n}^{\text{gausshyb}}(\mathbf{x}, \mathbf{y}))$ becomes zero in Theorem F.1, therefore the MSE remains the same. We thus obtain the second part of the theorem and that completes the proof.

$\square$

Now let us assume the setting of the same-lengths inputs. We denote the length by $r$ and an angle between inputs by $\theta$. We can rewrite Eq. 67 and provide equivalent definition for $\lambda$ by replacing $\mathbf{x}$ and $\mathbf{y}$ with their angle $\theta$ and norm $r$:

$$\lambda(\theta, r) = \frac{1 - \exp(-2\sigma^2 r^2 \sin(\frac{\theta}{2})^2)}{\rho} \tag{76}$$

We obtain the following version of the above theorem:

**Theorem F.2** (MSE of the normalized bipolar Gaussian hybrid estimator for same-length inputs).
*Assume that $\|\mathbf{x}\|_2 = \|\mathbf{y}\|_2 = r$ and denote: $\theta = \theta_{\mathbf{x},\mathbf{y}}$. Take the normalized bipolar Gaussian hybrid estimator $\widehat{\text{SM}}_{m,n}^{\text{gausshyb}}(\theta, r)$, where $\widehat{\text{SM}}_m^{\text{trig}}(\theta, r)$ and $\widehat{\text{SM}}_m^{++}(\theta, r)$ are chosen independently i.e. their random projections are chosen independently (note that we always assume that $\widehat{\lambda}(\theta, r)$ is chosen independently from $\widehat{\text{SM}}_m^{\text{trig}}(\theta, r)$ and $\widehat{\text{SM}}_m^{++}(\theta, r)$). Then the following holds:*

$$\text{MSE}(\widehat{\text{SM}}_{m,n}^{\text{gausshyb}}(\theta, r)) = \mathbb{E}[\widehat{\lambda}^2(\theta, r)]\text{MSE}(\widehat{\text{SM}}_m^{++}(\theta, r)) + \mathbb{E}[(1 - \widehat{\lambda}(\theta, r))^2]\text{MSE}(\widehat{\text{SM}}_m^{\text{trig}}(\theta, r)) \tag{77}$$

*where*

$$\mathbb{E}[(\widehat{\lambda}(\theta, r))^2] = \frac{1}{\rho^2}(1 - \exp(-2\sigma^2 r^2 \sin(\frac{\theta}{2})^2))^2 + \frac{1}{2\rho^2 n}(1 - \exp(-4\sigma^2 r^2 \sin(\frac{\theta}{2})^2))^2 \tag{78}$$

$$\mathbb{E}[(1 - \widehat{\lambda}(\theta, r))^2] = \frac{1}{\rho^2}((1-\rho) - \exp(-2\sigma^2 r^2 \sin(\frac{\theta}{2})^2))^2 + \frac{1}{2\rho^2 n}(1 - \exp(-4\sigma^2 r^2 \sin(\frac{\theta}{2})^2))^2 \tag{79}$$

*Furthermore, if $\widehat{\text{SM}}_m^{\text{trig}}(\theta, r)$ and $\widehat{\text{SM}}_m^{++}(\theta, r)$ apply the **exact** same sets of random projections, the mean squared error of the hybrid estimator will remains the same.*

*Proof.* For $\|\mathbf{x}\|^2 = \|\mathbf{y}\|^2 = r^2$, the following holds for $\Delta = \mathbf{x} - \mathbf{y}$:

$$\|\Delta\|^2 = 4r^2 \sin(\frac{\theta}{2})^2 \tag{80}$$

And this theorem holds trivially by replacing $\|\Delta\|^2$ with the above formula in Theorem F.1.

$\square$

# G  POINTWISE SOFTMAX KERNEL ESTIMATION EXPERIMENTS

In Fig. 5 we present complete ablation studies regarding empirical relative errors of different RF-based estimators of the softmax kernel over different lengths of inputs $\mathbf{r}$ and angles $\theta_{\mathbf{x},\mathbf{y}}$.

# H  COMPLEX EXPONENTIAL ESTIMATORS FOR CLUSTERED DATA

## H.1  SYNTHETIC EXPERIMENTS ON DATA ADMITTING CLUSTERING STRUCTURE

Here we test HRF estimators customized to data with clustering structure, as described in Sec. 2.4. Inputs $\mathbf{x}$ are taken from two 50-dimensional 1000-element Gaussian clusters and the inputs $\mathbf{y}$ from two other 50-dimensional 1000-element Gaussian clusters. We use empirical MSE metric and constrain matrix $\mathbf{A}$ to be real diagonal for a compact closed-form formulae (see: Sec. H.3). We created four different clusters' configurations corresponding to different values of $s = l(\mathbf{A})$ (see: Eq. 21) for all pairs of clusters and with different concentrations around centers controlled by the standard deviation $\sigma$ (small/large values of $s$ and $\sigma$). As shown in Fig. 6, for all variants HRF estimators adapted to clustered data consistently provide notable accuracy gains, even for larger values of $s$ and less concentrated clusters. Additional experimental details are given in Sec. H.3.

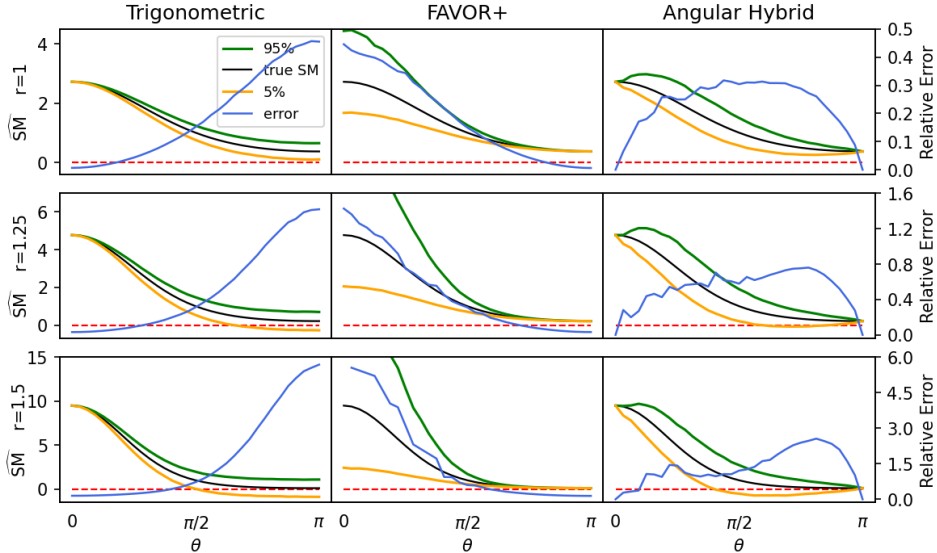

Figure 5: Pointwise estimation of $\mathrm{SM}(\mathbf{x}, \mathbf{y})$ for the same-length 64-dim inputs ($r = 1.0$, $r = 1.25$ and $r = 1.5$) and various angles $\theta_{\mathbf{x},\mathbf{y}}$. We used $s = 10000$ estimated softmax values in each subplot. The true softmax value and the $5^{th}$ and $95^{th}$ quantile estimated values are shown by the left y-axis, and the empirical relative errors are shown by the right y-axis. Trigonometric estimator and FAVOR+ applied 128 random features. To make fair comparison, for the hybrid variant the configuration leading to the similar number of FLOPS operations per random feature map creation was applied.

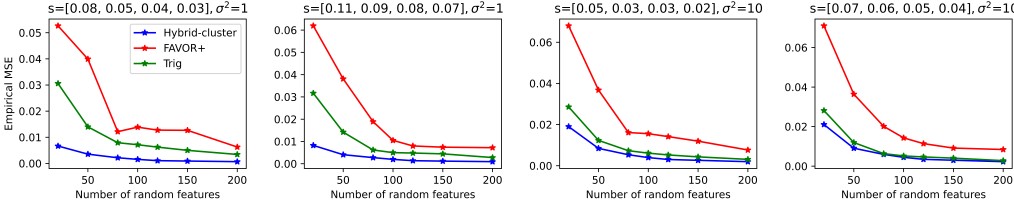

Figure 6: Comparing different estimators for data with clustering structure. The empirical MSE is obtained by averaging over 20 trials. For the HRFs, we apply deterministic $\lambda$-coefficients. The fact that the empirical error for FAVOR+ is not perfectly monotonic in $m$ was first observed in (Luo et al., 2021) (see: Fig. 1: (b)).

## H.2 INSTANTIATING COMPLEX EXPONENTIAL ESTIMATORS FOR CLUSTERED DATA

Assume that the inputs $\mathbf{x}$ (queries) can be modeled by $n_q$ clusters and that the inputs $\mathbf{y}$ (keys) can be modeled by $n_k$ clusters (e.g. via k-means clustering algorithm). Denote the center of each cluster as $\mathbf{r}_i \in \mathbb{R}^d$, ($i = 1, ..., n_k$) and $\mathbf{r}_j \in \mathbb{R}^d$, ($j = 1, ..., n_q$). Then there exist $n_q n_k$ pairs of $(\mathbf{r}_i, \mathbf{r}_j)$, ($i = 1, ..., n_q, j = 1, ..., n_k$), so we can construct $n_q n_k$ softmax kernel estimators to estimate cross-group softmax kernel values.

Consider $\mathbf{z} = \mathbf{A}\mathbf{r}_i + (\mathbf{A}^\top)^{-1}\mathbf{r}_j$ for an invertible (in $\mathbb{C}^{d \times d}$) matrix $\mathbf{A} \in \mathbb{C}^{d \times d}$. An estimator based on this mechanism has variance equal to $0$ if:

$$\mathbf{z} = \mathbf{A}\mathbf{r}_i + (\mathbf{A}^\top)^{-1}\mathbf{r}_j = 0 \tag{81}$$

From now on we constrain $\mathbf{A}$ to be diagonal, so $\mathbf{A} = \mathbf{A}^\top$. We can rewrite the above equation as:

$$\mathbf{z} = \mathbf{A}\mathbf{r}_i + (\mathbf{A})^{-1}\mathbf{r}_j = 0 \tag{82}$$

Since position $(k, k)$ in matrix $\mathbf{A}$ is a complex number of the form $\alpha_k + \beta_k i$, we need to satisfy the following equation for each $k = 1, ..., d$:

$$(\alpha_k + \beta_k i)r_{i,k} + \frac{1}{\alpha_k + \beta_k i}r_{j,k} = 0 \tag{83}$$

$$(\alpha_k + \beta_k i)r_{i,k} + \frac{\alpha_k - \beta_k i}{\alpha_k^2 + \beta_k^2}r_{j,k} = 0 \tag{84}$$

where $r_{i,k}$ is the $k$-th entry of vector $\mathbf{r}_i$.

We can simplify this equation and separate into real and imaginary part:

$$\begin{cases} \mathbf{Re} : (\alpha_k^2 + \beta_k^2)\alpha_k r_{i,k} + \alpha_k r_{j,k} = 0 \\ \mathbf{Im} : (\alpha_k^2 + \beta_k^2)\beta_k r_{i,k} - \beta_k r_{j,k} = 0 \end{cases} \tag{85}$$

Our goal now is to choose values for $\alpha_k$ and $\beta_k$ given $\mathbf{r}_i$ and $\mathbf{r}_j$.

If $r_{i,k} r_{j,k} > 0$, we can set:

$$\begin{cases} \alpha_k = 0 \\ \beta_k = \sqrt{r_{j,k}/r_{i,k}} \end{cases} \tag{86}$$

And if $r_{i,k} r_{j,k} < 0$, we can set:

$$\begin{cases} \alpha_k = \sqrt{-r_{j,k}/r_{i,k}} \\ \beta_k = 0 \end{cases} \tag{87}$$

When $r_{i,k} = r_{j,k} = 0$, we can take $\alpha_k = \beta_k = 1$. If $r_{i,k} = 0, r_{j,k} \neq 0$ or the opposite, then we cannot satisfy the above equation perfectly. We can take $\alpha_k$ to some large positive value and set $\beta_k = 0$ when $r_{i,k} = 0, r_{j,k} \neq 0$, and set $\alpha_k = \beta_k$ to some small positive value close to zero when $r_{i,k} \neq 0, r_{j,k} = 0$.

When restricting $\mathbf{A} = \mathrm{diag}(a_1, \ldots, a_d)$ to $\mathbb{R}^{d \times d}$, it is easily seen that to minimize $|\mathbf{A}\mathbf{r}_i + (\mathbf{A}^\top)^{-1}\mathbf{r}_j|^2$, we can take

$$a_k = \sqrt{|r_{j,k}/r_{i,k}|} \tag{88}$$

if $r_{i,k} \neq 0$. And if $r_{i,k} = 0, r_{j,k} \neq 0$, $a_k$ can be set to a large positive number. We can set $a_k = 1$ if $r_{i,k} = 0, r_{j,k} = 0$.

In the analysis below we denote matrix $\mathbf{A}$ calculated from Eq. 85 and Eq. 88 (depending on whether we are using real or complex matrices) given $\mathbf{r}_i$ and $\mathbf{r}_j$ as $\mathbf{A}^{i,j}$.

From the analysis in the main body of the paper we know that for arbitrary vectors $\mathbf{x}, \mathbf{y} \in \mathbb{R}^d$:

$$\exp\left(\frac{\|\mathbf{A}^{i,j}\mathbf{x}\|^2}{2}\right)\exp\left(\frac{\|(\mathbf{A}^{i,j})^{-1}\mathbf{y}\|^2}{2}\right)\mathrm{SM}(\mathbf{x},\mathbf{y}) = \mathbb{E}[\exp(\omega^\top(\mathbf{A}^{i,j}\mathbf{x} + (\mathbf{A}^{i,j})^{-1}\mathbf{y})] \tag{89}$$

Therefore, the softmax kernel can be estimated as:

$$\mathrm{SM}(\mathbf{x},\mathbf{y}) \approx \Psi_{\mathbf{A}^{i,j}}^m(\mathbf{x})^\top \Psi_{(\mathbf{A}^{i,j})^{-1}}^m(\mathbf{y}) \tag{90}$$

with $\Psi_{\mathbf{A}^{i,j}}^m(\mathbf{x})$ and $\Psi_{(\mathbf{A}^{i,j})^{-1}}^m(\mathbf{y})$ given by:

$$\Psi_{\mathbf{A}^{i,j}}^m(\mathbf{x}) \stackrel{\text{def}}{=} \frac{1}{\sqrt{m}}\exp\left(-\frac{\|\mathbf{A}^{i,j}\mathbf{x}\|^2}{2}\right)(\exp(\omega_1^\top \mathbf{A}^{i,j}\mathbf{x}), \ldots, \exp(\omega_m^\top \mathbf{A}^{i,j}\mathbf{x}))^\top \tag{91}$$

$$\Psi_{(\mathbf{A}^{i,j})^{-1}}^m(\mathbf{y}) \stackrel{\text{def}}{=} \frac{1}{\sqrt{m}}\exp\left(-\frac{\|(\mathbf{A}^{i,j})^{-1}\mathbf{y}\|^2}{2}\right)(\exp(\omega_1^\top(\mathbf{A}^{i,j})^{-1}\mathbf{y}), \ldots, \exp(\omega_m^\top(\mathbf{A}^{i,j})^{-1}\mathbf{y}))^\top \tag{92}$$

for $\omega_1, \ldots, \omega_m \sim \mathcal{N}(0, \mathbf{I}_d)$.

Such a softmax kernel estimator has variance equal to zero if $\mathbf{x} = \mathbf{r}_i$ and $\mathbf{y} = \mathbf{r}_j$ if we are using the complex matrix and has small variance if we are using the real matrix. For other pairs of vectors $(\mathbf{x}, \mathbf{y})$ close to $\mathbf{r}_i$ and $\mathbf{r}_j$, the variance is close to zero, or relatively small.

We can also take the Gaussian $\lambda$-coefficient estimator that gets it maximum value when $\mathbf{x} = \mathbf{r}_i, \mathbf{y} = \mathbf{r}_j$ as $\lambda^{i,j}(\mathbf{x}, \mathbf{y})$. Such an estimator could be constructed with $\mathbf{A}^{i,j}$ in a similar way.

It can be easily linearized since:

$$\lambda^{i,j}(\mathbf{x}, \mathbf{y}) = \exp(-\frac{\|\mathbf{A}^{i,j}\mathbf{x} + (\mathbf{A}^{i,j})^{-1}\mathbf{y}\|^2}{2\tau^2}) =$$
$$\exp(-\frac{\|\mathbf{A}^{i,j}\mathbf{x}\|^2}{2\tau^2})\exp(-\frac{\|(\mathbf{A}^{i,j})^{-1}\mathbf{y}\|^2}{2\tau^2})\mathrm{SM}(\frac{\mathbf{x}}{\tau}, \frac{-\mathbf{y}}{\tau}), \tag{93}$$

where $\mathrm{SM}(\mathbf{x}/\tau, -\mathbf{y}/\tau)$ could be estimated with similar random feature maps as given by Eq. 90. Therefore, the Gaussian $\lambda$-coefficients can be estimated as:

$$\lambda^{i,j}(\mathbf{x}, \mathbf{y}) = \exp(-\frac{\|\mathbf{A}^{i,j}\mathbf{x} + (\mathbf{A}^{i,j})^{-1}\mathbf{y}\|^2}{2\tau^2}) \approx \Psi_{\mathbf{A}^{i,j}}^{\tau,m}(\mathbf{x})^\top \Psi_{(\mathbf{A}^{i,j})^{-1}}^{\tau,m}(\mathbf{y}) \tag{94}$$

with $\Psi_{\mathbf{A}^{i,j}}^{\tau,m}(\mathbf{x})$ and $\Psi_{(\mathbf{A}^{i,j})^{-1}}^{\tau,m}(-\mathbf{y})$ given by:

$$\Psi_{\mathbf{A}^{i,j}}^{\tau,m}(\mathbf{x}) = \frac{1}{\sqrt{m}}\exp(-\frac{\|\mathbf{A}^{i,j}\mathbf{x}\|^2}{\tau^2})(\exp(\omega_1^\top \mathbf{A}^{i,j}\mathbf{x}/\tau), ..., \exp(\omega_m^\top \mathbf{A}^{i,j}\mathbf{x}/\tau))^\top \tag{95}$$

$$\Psi_{(\mathbf{A}^{i,j})^{-1}}^{\tau,m}(-\mathbf{y}) = \frac{1}{\sqrt{m}}\exp(-\frac{\|(\mathbf{A}^{i,j})^{-1}\mathbf{y}\|^2}{\tau^2})(\exp(-\omega_1^\top (\mathbf{A}^{i,j})^{-1}\mathbf{y}/\tau), ..., \exp(-\omega_m^\top (\mathbf{A}^{i,j})^{-1}\mathbf{y}/\tau))^\top \tag{96}$$

for $\omega_1, ..., \omega_m \sim \mathcal{N}(0, \mathbf{I}_d)$ chosen independently and that were applied in base estimators.

We summarize this section with the following two constructions of the hybrid estimators for the clustered data that are implied by the above analysis.

**Definition H.1** (Hybrid Gaussian-Mixtures estimators for clustered data). *Assume that inputs $\mathbf{x}$ (queries) and $\mathbf{y}$ (keys) can be modeled by $n_q$ and $n_k$ clusters respectively with centers $\mathbf{r}_i$ and $\mathbf{r}_j \in \mathbb{R}^d$, ($i = 1, ..., n_k$, $j = 1, ..., n_q$). We denote $\mathbf{A}^{i,j}$ as the complex matrix satisfying Eq. (85) with center $\mathbf{r}_i, \mathbf{r}_j$. Furthermore, we denote $\mathcal{E} = (\widehat{\mathrm{SM}}^{i,j}(\mathbf{x}, \mathbf{y}))_{i=1,j=1}^{n_q,n_k}$ as a list of estimators of the softmax kernel $\mathrm{SM}^{i,j}(\mathbf{x}, \mathbf{y})$ and $\Lambda = \widehat{\lambda}^{i,j}(\mathbf{x}, \mathbf{y}))_{i=1,j=1}^{n_q,n_k}$ as a list of estimators of $\lambda^{i,j}(\mathbf{x}, \mathbf{y})$ constructed independently from $\mathcal{E}$. We also use one additional base estimator (e.g. $\widehat{\mathrm{SM}}^{\mathrm{trig}}(\mathbf{x}, \mathbf{y})$ or $\widehat{\mathrm{SM}}^{++}(\mathbf{x}, \mathbf{y})$) and denote the estimator of its softmax kernel as $\widehat{\mathrm{SM}}^0(\mathbf{x}, \mathbf{y})$ and $\lambda$-coefficient as $\widehat{\lambda}^0(\mathbf{x}, \mathbf{y})$. Then our hybrid estimator takes the following form:*

$$\widehat{\mathrm{SM}}^{\mathcal{E},\Lambda}(\mathbf{x}, \mathbf{y}) = \sum_{i=1}^{n_q}\sum_{j=1}^{n_k}\widehat{\lambda}^{i,j}(\mathbf{x}, \mathbf{y})\widehat{\mathrm{SM}}^{i,j}(\mathbf{x}, \mathbf{y}) + \widehat{\lambda}^0(\mathbf{x}, \mathbf{y})\widehat{\mathrm{SM}}^0(\mathbf{x}, \mathbf{y}) \tag{97}$$

*with constraint:*

$$\sum_{i=1}^{n_q}\sum_{j=1}^{n_k}\widehat{\lambda}^{i,j}(\mathbf{x}, \mathbf{y}) + \widehat{\lambda}^0(\mathbf{x}, \mathbf{y}) = 1 \tag{98}$$

*and where the estimators of $\lambda$-coefficients and base estimators are given as:*

$$\widehat{\lambda}^{i,j}(\mathbf{x}, \mathbf{y}) = \Psi_{\mathbf{A}^{i,j}}^{\tau,m}(\mathbf{x})^\top \Psi_{(\mathbf{A}^{i,j})^{-1}}^{\tau,m}(-\mathbf{y}) \tag{99}$$

$$\widehat{\mathrm{SM}}^{i,j}(\mathbf{x}, \mathbf{y}) = \Psi_{\mathbf{A}^{i,j}}^m(\mathbf{x})^\top \Psi_{(\mathbf{A}^{i,j})^{-1}}^m(\mathbf{y}) \tag{100}$$

*for $\Psi_{\mathbf{A}^{i,j}}^m(\mathbf{x}), \Psi_{(\mathbf{A}^{i,j})^{-1}}^m(\mathbf{y}), \Psi_{\mathbf{A}^{i,j}}^{\tau,m}(\mathbf{x}), \Psi_{(\mathbf{A}^{i,j})^{-1}}^{\tau,m}(-\mathbf{y})$ given by Eq. (91, 92, 95, 96).*

**Definition H.2** (Hybrid Zero-One-Mixtures estimators for clustered data). *Assume that inputs* $\mathbf{x}$ *(queries) and* $\mathbf{y}$ *(keys) can be modeled by* $n_q$ *and* $n_k$ *clusters respectively with centers* $\mathbf{r}_i$ *and* $\mathbf{r}_j \in \mathbb{R}^d$, $(i = 1, ..., n_k, \; j = 1, ..., n_q)$. *We denote* $\mathbf{A}^{i,j}$ *as the complex matrix satisfying Eq. (85) with center* $\mathbf{r}_i, \mathbf{r}_j$. *Furthermore, we denote* $\mathcal{E} = (\widehat{\mathrm{SM}}^{i,j}(\mathbf{x}, \mathbf{y}))_{i=1, j=1}^{n_q, n_k}$ *as a list of estimators of the softmax kernel* $\mathrm{SM}^{i,j}(\mathbf{x}, \mathbf{y})$ *and* $\Lambda = \widehat{\lambda}^{i,j}(\mathbf{x}, \mathbf{y}))_{i=1, j=1}^{n_q, n_k}$ *as a list of estimators of* $\lambda^{i,j}(\mathbf{x}, \mathbf{y})$ *constructed independently from* $\mathcal{E}$.

*Then our hybrid estimator takes the following form:*

$$\widehat{\mathrm{SM}}^{\mathcal{E}, \Lambda}(\mathbf{x}, \mathbf{y}) = \sum_{i=1}^{n_q} \sum_{j=1}^{n_k} \widehat{\lambda}^{i,j}(\mathbf{x}, \mathbf{y}) \widehat{\mathrm{SM}}^{i,j}(\mathbf{x}, \mathbf{y}) \tag{101}$$

*with constraint:*

$$\sum_{i=1}^{n_q} \sum_{j=1}^{n_k} \widehat{\lambda}^{i,j}(\mathbf{x}, \mathbf{y}) = 1 \tag{102}$$

*and where the estimators of* $\lambda$-*coefficients and base estimators are given as:*

$$\widehat{\lambda}^{i,j}(\mathbf{x}, \mathbf{y}) = \Psi_i(\mathbf{x}) \Psi_j(\mathbf{y}) \tag{103}$$

$$\widehat{\mathrm{SM}}^{i,j}(\mathbf{x}, \mathbf{y}) = \Psi_{\mathbf{A}^{i,j}}^m(\mathbf{x})^\top \Psi_{(\mathbf{A}^{i,j})^{-1}}^m(\mathbf{y}) \tag{104}$$

*for* $\Psi_{\mathbf{A}^{i,j}}^m(\mathbf{x}), \Psi_{(\mathbf{A}^{i,j})^{-1}}^m(\mathbf{y})$ *given by Eq. (91, 92), with* $\Psi_i(\mathbf{x})$ *being a scalar indicating whether* $\mathbf{x}$ *belongs to the* $i$-*th cluster of the* $n_q$ *clusters and similarly with* $\Psi_j(\mathbf{y})$ *being a scalar indicating whether* $\mathbf{y}$ *belongs to the* $j$-*th of the* $n_k$ *clusters.*

In our experiments we used the formulation from the second definition with $\mathbf{A}$ constrained to be real diagonal. In other words, we are using Eq. 88 to construct $\mathbf{A}$.

## H.3 Additional experimental details

To create the synthetic data, we first generate two random vectors $X_0, Y_0$ in $\mathbb{R}^{50}$. Then we generate four random orthogonal matrices $O^{i,j} \in \mathbb{R}^{50 \times 50}$ where $i = 1, 2$ and $j = 1, 2$ and use $X_i = O^{i,1} X_0$ and $Y_i = O^{i,2} Y_0$ for $i = 1, 2$ as the mean vectors for our Gaussian clusters. For each pair of clusters, the minimal value $s$ of Eq. 21 may be non-zero if $\mathrm{sign}(\mathbf{x}_{i,k}) = \mathrm{sign}(\mathbf{y}_{j,k})$ for some $k$ due to the usage of real diagonal matrices. To control the value of $s$, we manually adjust the sign of each element of $X_i$ and $Y_i$. The data for input $\mathbf{x}$ is the combination of two clusters, each consisting 1000 data points following Gaussian distribution with mean vectors $X_1, X_2$ and the common covariance matrix $\sigma^2 I$. And similar constructions are done for input $\mathbf{y}$. We create four synthetic data sets, with different values of $s$ and magnitudes of the standard deviation $\sigma$. After this step, we normalize the data points by controlling their $\ell_2$ norms to make the true softmax value be within a reasonable range. After the data is created, we first use K-means clustering algorithm with $k = 2$ to cluster $\mathbf{x}$ and $\mathbf{y}$, and we obtain centers $(\mathbf{x}_i, \mathbf{y}_j)_{i,j \in \{1,2\}}$. Then we use the set of real diagonal matrices to minimize Eq. 21 for all pairs of centers of $\mathbf{x}$ and $\mathbf{y}$ to generate four complex exponential base estimators. To choose $\lambda$ coefficients, we use the more efficient approach described above, i.e. using indicator vectors, and these are already computed in the clustering process. For each data set, we compare the performance of these estimators by calculating the mean square error of them, and we do this multiple times with different numbers of random features used. All the results are obtained by averaging over 20 repetitions. In addition to the average performance which can be found in Fig. 6, we also record the maximal and minimal empirical MSE over 20 repetitions for all estimators in the Table 3, where $s_1 = [0.08, 0.05, 0.04, 0.03], s_2 = [0.11, 0.09, 0.08, 0.07], s_3 = [0.05, 0.03, 0.03, 0.02], s_4 = [0.07, 0.06, 0.05, 0.04]$ representing the lists of $s$ values for all pairs of clusters for different synthetic data sets.

| Data Set | Number of Random Features | Hybrid-Cluster | FAVOR+ | Trigonometric |
|---|---|---|---|---|
| $s = s_1, \sigma = 1$ | 20 | [0.0016, 0.0183] | [0.0071, 0.1455] | [0.0039, 0.0808] |
| | 50 | [0.0011, 0.0081] | [0.0026, 0.2541] | [0.0031, 0.0510] |
| | 80 | [0.0005, 0.0052] | [0.0032, 0.0282] | [0.0010, 0.0281] |
| | 100 | [0.0006, 0.0036] | [0.0022, 0.0528] | [0.0013, 0.0170] |
| | 120 | [0.0004, 0.0025] | [0.0028, 0.0534] | [0.0012, 0.0135] |
| | 150 | [0.0003, 0.0020] | [0.0014, 0.0582] | [0.0007, 0.0134] |
| | 200 | [0.0002, 0.0010] | [0.0013, 0.0205] | [0.0004, 0.0087] |
| $s = s_2, \sigma = 1$ | 20 | [0.0032, 0.0265] | [0.0095, 0.2071] | [0.0036, 0.0989] |
| | 50 | [0.0012, 0.0079] | [0.0036, 0.2123] | [0.0019, 0.0565] |
| | 80 | [0.0007, 0.0059] | [0.0037, 0.0570] | [0.0008, 0.0208] |
| | 100 | [0.0007, 0.0040] | [0.0016, 0.0280] | [0.0010, 0.0164] |
| | 120 | [0.0005, 0.0025] | [0.0013, 0.0262] | [0.0007, 0.0246] |
| | 150 | [0.0004, 0.0024] | [0.0010, 0.0407] | [0.0004, 0.0190] |
| | 200 | [0.0003, 0.0018] | [0.0018, 0.0236] | [0.0004, 0.0134] |
| $s = s_3, \sigma = \sqrt{10}$ | 20 | [0.0083, 0.0300] | [0.0362, 0.1218] | [0.0174, 0.0490] |
| | 50 | [0.0046, 0.0183] | [0.0125, 0.1262] | [0.0081, 0.0249] |
| | 80 | [0.0029, 0.0105] | [0.0108, 0.0281] | [0.0044, 0.0143] |
| | 100 | [0.0024, 0.0056] | [0.0077, 0.0421] | [0.0041, 0.0095] |
| | 120 | [0.0020, 0.0046] | [0.0077, 0.0402] | [0.0034, 0.0077] |
| | 150 | [0.0015, 0.0041] | [0.0049, 0.0395] | [0.0025, 0.0073] |
| | 200 | [0.0011, 0.0031] | [0.0037, 0.0150] | [0.0018, 0.0050] |
| $s = s_4, \sigma = \sqrt{10}$ | 20 | [0.0097, 0.0294] | [0.0332, 0.1553] | [0.0179, 0.0485] |
| | 50 | [0.0049, 0.0172] | [0.0176, 0.1345] | [0.0058, 0.0260] |
| | 80 | [0.0031, 0.0113] | [0.0113, 0.0415] | [0.0038, 0.0110] |
| | 100 | [0.0028, 0.0061] | [0.0080, 0.0292] | [0.0031, 0.0100] |
| | 120 | [0.0022, 0.0045] | [0.0065, 0.0185] | [0.0028, 0.0128] |
| | 150 | [0.0016, 0.0044] | [0.0045, 0.0235] | [0.0021, 0.0094] |
| | 200 | [0.0012, 0.0031] | [0.0046, 0.0166] | [0.0015, 0.0067] |

Table 3: Minimal and maximal empirical MSE with 20 repetitions

# I LANGUAGE MODELING TRAINING DETAILS AND ADDITIONAL RESULTS

For the Language Modeling tasks, we trained a 2-layer LSTM of hidden sizes 200 and 650 on the PennTree Bank (Marcus et al., 1993) and the WikiText2 dataset (Merity et al., 2017) respectively. We tied the weights of the word embedding layer and the decoder layer (Inan et al., 2017; Press & Wolf, 2017). Thus we could treat the language modeling problem as minimizing the cross entropy loss between the dot product of the model output (queries) and the class embedding (keys) obtained from the embedding layer with the target word.

We now present detailed statistics of HRFs on the Penn Tree Bank dataset. The mean and the standard deviation of the statistical metric results in Fig. 3, Sec. 4.2 is shown in Table 4. The distribution of the lengths of the keys and the queries are shown in Figure 7. For our experiments with the Angular Hybrid and the Gaussian Hybrid, the number of random features are $8, 16, 32, 64$ for the base estimators, and $8$ for the coefficient estimators.

Table 4: Results are computed over 10 runs on Penntree Bank Dataset. **Boldface** denotes the best one, and underline denotes the second best. Negative fractions for Favor+ is not reported as it produces positive random features.

| RF types | $d = 64$ | $d = 128$ | $d = 256$ | $d = 512$ |
|---|---|---|---|---|
| $1d$-Wasserstein | | | | |
| FAVOR+ | $3171.35 \pm 69.04$ | $3142.65 \pm 85.55$ | $3050.58 \pm 108.25$ | $2985.18 \pm 73.15$ |
| Trig. | $1577.44 \pm 13.59$ | $1582.50 \pm 13.42$ | $1574.22 \pm 7.01$ | $1576.50 \pm 5.83$ |
| Gaussian | $1520.32 \pm 11.65$ | $1521.10 \pm 8.79$ | $1516.50 \pm 6.66$ | $1515.41 \pm 5.96$ |
| Angular | $\mathbf{1395.03 \pm 79.91}$ | $\mathbf{1445.24 \pm 58.27}$ | $\mathbf{1394.97 \pm 62.73}$ | $\mathbf{1425.88 \pm 65.82}$ |
| KS statistics | | | | |
| FAVOR+ | $0.573 \pm 0.0073$ | $0.569 \pm 0.0098$ | $0.558 \pm 0.013$ | $0.551 \pm 0.0092$ |
| Trig. | $0.424 \pm 0.0040$ | $0.427 \pm 0.0041$ | $0.425 \pm 0.0024$ | $0.424 \pm 0.0023$ |
| Gaussian | $\underline{0.411 \pm 0.0029}$ | $\mathbf{0.410 \pm 0.0019}$ | $\mathbf{0.410 \pm 0.0013}$ | $\mathbf{0.410 \pm 0.0014}$ |
| Angular | $\mathbf{0.404 \pm 0.017}$ | $\underline{0.419 \pm 0.010}$ | $\underline{0.418 \pm 0.0089}$ | $\underline{0.414 \pm 0.0093}$ |
| Negative fraction | | | | |
| Trig. | $0.257 \pm 0.0057$ | $0.258 \pm 0.0039$ | $0.258 \pm 0.0021$ | $0.257 \pm 0.0017$ |
| Gaussian | $0.155 \pm 0.0039$ | $\underline{0.159 \pm 0.0019}$ | $\underline{0.157 \pm 0.0011}$ | $\mathbf{0.151 \pm 0.0009}$ |
| Angular | $\mathbf{0.152 \pm 0.0041}$ | $\mathbf{0.158 \pm 0.0027}$ | $\mathbf{0.153 \pm 0.0018}$ | $\underline{0.157 \pm 0.0013}$ |

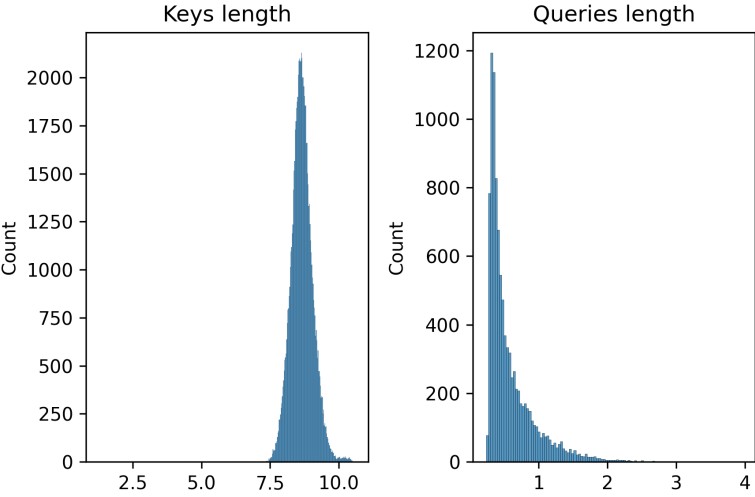

Figure 7: Distribution of the lengths of the keys and queries in the PennTree Bank dataset

Since the HRF mechanisms can be sensitive to the lengths of the keys and the queries, for the language modeling task on the WikiText2 dataset, we added a regularizer to constrain the lengths of the keys and the queries to be close to 1. However, constraining the lengths close to 1 hurts the model performance and so we chose to use a temperature scaling as in (Rawat et al., 2019). Thus before passing the computed dot product to the softmax layer, we scaled the dot product by $\tau > 1$. Our final loss function for the language modeling task was:

$$\mathcal{L}(\theta) = L_{CE} + \lambda_1 \mathbb{E}(||q||_2 - \mathbf{1})^2 + \lambda_2 \mathbb{E}(||k||_2 - \mathbf{1})^2 \tag{105}$$

where $|| \cdot ||_2$ is the row-wise $L_2$ norm of the appropriate matrices, and $L_{CE}$ is the cross-entropy loss. The distribution of the lengths of the keys and the queries are shown in Figure 9.

Finally we present some additional results on the WikiText2 dataset. For our experiments on the WikiText2 dataset, we chose $\lambda_1 = \lambda_2 = 2$ and $\tau = 6$. Our model trained with these hyperparameters achieve a perplexity score of 105.35 on the test set after 50 epochs.

We compute the 1-dimensional Wasserstein distance and the Kolmogrov-Smirnov (KS) metric (Ramdas et al., 2017) between the approximated softmax distribution and the true softmax distribution. The mean and the standard deviation of the statistical metric results in Fig. 8 is shown in Table 5. For our experiments with the Angular Hybrid and the Gaussian Hybrid, the number of random features are $8, 16, 32, 64$ for the base estimators, and $8$ for the coefficient estimators.

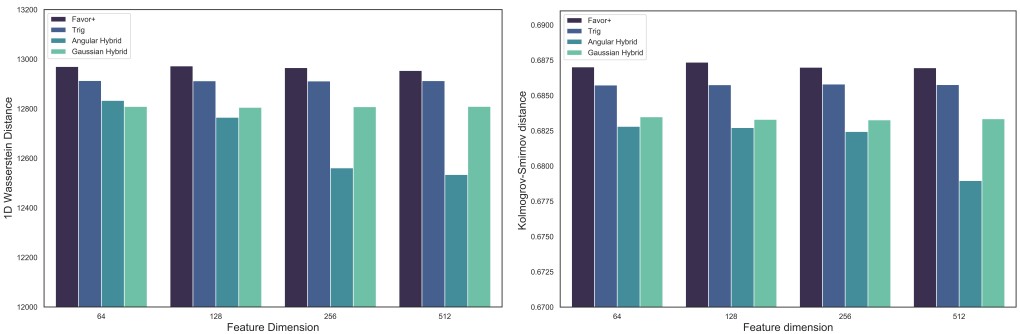

Figure 8: Statistical metrics measuring softmax matrix approximation quality on the WikiText2 dataset. For standard estimators, the number of random features are $64, 128, 256, 512$. To make fair comparison, for the hybrid variants, the configurations leading to the similar number of FLOPS operations per random feature map creation were applied. Results reported over 10 runs.

Table 5: Results are computed over 10 runs on the Wikitext2 Dataset. **Boldface** denotes the best one, and underline denotes the second best.

| RF types | $d = 64$ | $d = 128$ | $d = 256$ | $d = 512$ |
|---|---|---|---|---|
| | 1$d$-Wasserstein | | | |
| FAVOR+ | $12970.38 \pm 119.49$ | $12971.67 \pm 108.79$ | $12965.82 \pm 89.62$ | $12954.17 \pm 83.95$ |
| Trig. | $12913.66 \pm 59.22$ | $12912.65 \pm 57.42$ | $12910.91 \pm 37.23$ | $12913.25 \pm 35.12$ |
| Gaussian | $\mathbf{12809.03 \pm 97.2}$ | $\underline{12805.71 \pm 78.91}$ | $\underline{12808.02 \pm 84.57}$ | $\underline{12809.40 \pm 65.88}$ |
| Angular | $\underline{12833.56 \pm 41.87}$ | $\mathbf{12765.44 \pm 38.91}$ | $\mathbf{12561.26 \pm 26.48}$ | $\mathbf{12534.48 \pm 25.88}$ |
| | KS statistics | | | |
| FAVOR+ | $0.687 \pm 0.0043$ | $0.687 \pm 0.0051$ | $0.687 \pm 0.0019$ | $0.686 \pm 0.0067$ |
| Trig. | $0.685 \pm 0.0071$ | $0.685 \pm 0.0053$ | $0.685 \pm 0.0029$ | $0.685 \pm 0.0014$ |
| Gaussian | $\underline{0.683 \pm 0.0092}$ | $\underline{0.683 \pm 0.0073}$ | $\underline{0.683 \pm 0.0049}$ | $\underline{0.682 \pm 0.0089}$ |
| Angular | $\mathbf{0.682 \pm 0.0079}$ | $\mathbf{0.682 \pm 0.0066}$ | $\mathbf{0.682 \pm 0.0037}$ | $\mathbf{0.678 \pm 0.0064}$ |

The above tables (Table 4 and Table 5) show that the our hybrid variants consistently outperform Favor+ and the trigonometric random features.

## I.1 LANGUAGE MODELING USING HRF

In this subsection, we will describe how one can use HRF to train a LSTM on the language modeling task on the PennTree Bank dataset, similar to the experiments carried out in (Rawat et al., 2019).

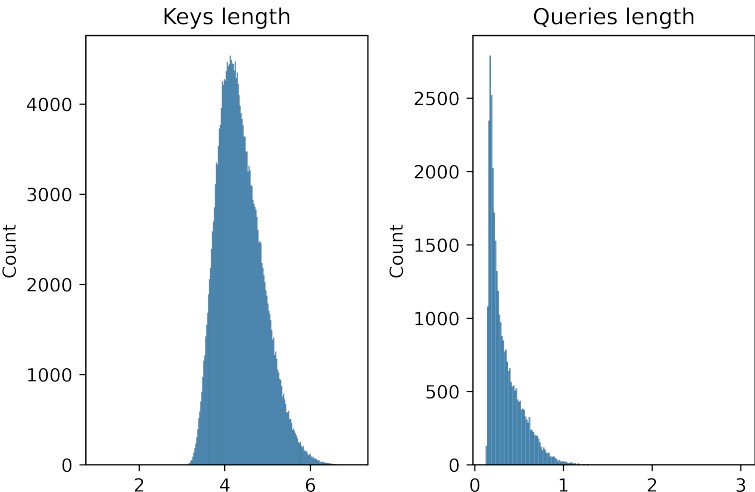

Figure 9: Distribution of the lengths of the keys and queries in the WikiText2 dataset

We trained a 2-layer LSTM model with hidden and output sizes of 200, and used the output as input embedding for sampled softmax. We sampled 40 negative (other than true) classes out of 10000 classes to approximate the expected loss in each training epoch. As observed in Fig. 2, the relative error of all three estimators grow exponentially fast with embedding norm $r$. Therefore, if we keep un-normalized embeddings during sampling, then even though we could get an unbiased estimation of the loss, the variance could be high. Such bias-variance trade off is also mentioned in paper (Rawat et al., 2019). To solve this issue, we used normalized input and class embeddings during sampling to generate biased sampling distribution with lower variance, while keeping un-normalized embeddings to calculate the loss. We trained our model for 80 epochs, with batch size equal to 20, dropout ratio in LSTM equal to 0.5. Implementation details could be seen in our Github repository.

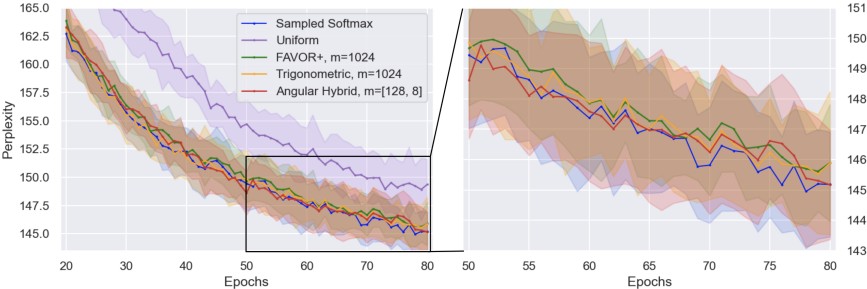

Figure 10: Language Modeling with softmax sampling: training results on PennTree Bank dataset (50-80 epoch window zoomed in on the right subfigure). The solid line for each method is estimated over 25 independent runs. The shaded areas represent perplexity within 1 standard deviation of the average. FA-VOR+/trigonometric mechanisms used 1024 random features. To make fair comparison, for HRFs, the configurations leading to the similar number of FLOPS operations per random feature map creation were applied. 128 and 8 random features is used to estimate the softmax estimators and $\lambda$-coefficient in HRFs respectively. Even though statistical metrics for HRFs are better in Fig. 3, the difference between different random feature estimators is not very significant.

We compared our hybrid estimator with Trigonometric, FAVOR+, the uniform method which sampled 40 negative classes with equal probability, as well as the sampled softmax method which uses the unbiased true probability calculated from full softmax as the sampling probability. Trigonometric and FAVOR+ mechanisms used 1024 random features. To make comparison fair, hybrid variants were using $(m, n)$-configurations characterized by the similar number of FLOPS needed to construct their corresponding random features as regular RF-estimators. We reported our comparison

results averaged over 25 independent runs in Fig. 10. The *perplexity* score in Fig. 10 is defined as $2^{\text{cross entropy loss}}$. We could conclude that even though the statistical metrics for HRFs are better in Fig. 3, the difference between HRFs and other random feature estimators in this specific softmax sampling downstream task is not very significant.

## J  Speech Experiments: Additional Details

Tested Conformer-Performer models consisted of $l = 17$ conformer layers. Each attention layer used $H = 8$ heads. The embedding dimensionality was $p = 512$ and since dimensions were split equally among different heads, query/key dimensionality was set up to $d_{\text{QK}} = 64$. Each input sequence was of length $L \approx 500$. Padding mechanism was applied for all tested variants.

## K  Downstream Robotics Experiments

In both Robotics experiments we found query/key $L_2$-normalization technique particularly effective (queries and keys of $L_2$-norm equal to one) thus we applied it in all the experiments.

### K.1  Visual Locomotion with Quadruped Robots

We evaluate HRF-based attention RL policies in a robotic task of learning locomotion on uneven terrains from vision input. We use the quadruped from Unitree called Laikago (lai). It has 12 actuated joints, 3 per leg. The task is to walk forward on a randomized uneven terrain that requires careful foot placement planning based on visual feedback. The ground is made of a series of step-stones with gaps in between. The step stones widths are fixed at $50$ cm, the lengths are between $[50, 80]$ cm in length, and the gap size between adjacent stones are between $[10, 20]$ cm. It perceives the ground through 2 depth cameras attached to its body, one on the front and other on the belly facing downwards. We use Implicit Attention Policy (IAP) architecture (masking variant) described in Choromanski et al. (2021a) which uses Performer-based attention mechanism to process $32 \times 24$ depth images from the 2 cameras.

We apply a hierarchical setup to solve this task. The high level uses IAP-rank with masking to process camera images and output the desired foot placement position. The low level employs a position-based swing leg controller, and an model predictive control (MPC) based stance leg controller, to achieve the foot placement decided by high level. The policies are trained with evolutionary strategies (ES). In Fig. 4 (left) we compare FAVOR+ with angular hybrid approximation in IAP. HRF with 8 x 8 random projections ($m = n = 8$) performs as well as the softmax kernel with 256 random projections. A series of image frames along the episode of a learned locomotion HRF-based IAP policy is shown bottom right of Fig. 4. On the top-left corner of the images, the input camera images are attached. The red part of the camera image is the area masked out by self-attention. The policy learns to pay attention to the gaps in the scene in order to avoid stepping on them.

The training curves in Fig. 4 (left) are obtained by averaging over 5 top runs. The shaded regions shows the standard deviation over multiple runs.

### K.2  Robotic Manipulation: Bi-manual Sweeping

Here we consider the bi-manual sweep robotic-arm manipulation. The two-robotic-arm system needs to solve the task of placing red balls into green bowls (all distributions are equally rewarded, so in particular robot can just use one bowl).

In this bi-manual sweeping task, the reward per step is between 0 and 1, the fraction of the blocks within the green bowls. The loss for this task is the negative log likelihood between the normalized softmax probability distribution of sampled actions (one positive example, and all others uniform negative counter-examples), and the ground truth one-hot probability distribution for a given observation from the training dataset. For more details see (van den Oord et al., 2018). This is a 12-DoF cartesian control problem, the state observation consists of the current cartesian pose of the arms and a $500 \times 500 \times 3$ RGB image. An example of the camera image is given in Fig. 11.

The training curves Fig. 4 (right) are obtained by averaging over 3 learning rates: $10^{-3}, 10^{-4}, 10^{-5}$.

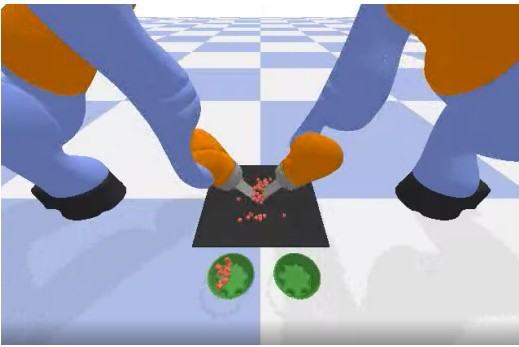

Figure 11: An example of the camera image for the bi-manual sweep robotic-arm manipulation task.

## L  COMPUTATIONAL GAINS OF HYBRID ESTIMATORS

Consider the general hybrid estimator of $\mathrm{SM}(\mathbf{x}, \mathbf{y})$ from Equation 8, using $p + 1$ base estimators. Assume that the $k^{th}$ base estimator for $k = 1, ..., p + 1$ applies $m$ random features in each of the corresponding $t_k$ random maps and that $n$ random features are used in each of the corresponding $l_{\lambda_k}$ random maps to approximate $k^{th}$ $\lambda$-coefficient for $k = 1, ..., p$. From Equation 11, we conclude that time complexity of constructing $\Psi(\mathbf{z})$ is:

$$T = O\left( (t_1 + ... + t_{p+1})md + (nd + md + mn) \sum_{k=1}^{p+1} \sum_{r=1}^{p} t_k l_r \right) \qquad (106)$$

Thus the resulting $\Theta(mn)$-dimensional random feature map $\Psi(\mathbf{z})$ can be constructed in time $O(nd + md + mn)$ as opposed to $O(mnd)$ as it is the case for the regular estimator, providing substantial computational gains. This is true if the $(\theta)(mn)$-dimensional HRFs can provide similar quality as their $mn$-dimensional regular counterparts. It turns out that this is often the case, see: Section D.4.

