# OpenReview forum: "Hybrid Random Features"
_ICLR.cc/2022/Conference — ICLR 2022 Poster_

### Official Review · Reviewer_Ks6B · 2021-10-28

**Correctness:** 3
**Technical Novelty And Significance:** 3
**Empirical Novelty And Significance:** 3
**Recommendation:** 8
**Confidence:** 5

**Main Review:**

Pros:

1, The motivation of this work is clear: current random features based algorithms are difficult to approximate softmax kernels with very small/large values. I agree with the authors on this issue.

2, the algorithm to approximate softmax kernel in Sec. 2.3.1 can be regarded as a combination of SM^{++} and SM^{trig}, with an extra cosine similarity function.
This function is in essence close to the zero-order arc-cosine kernel [Cho and Saul, 2009], and thus is actually approximated by the standard random features with the Hesivide activiation function. The algorithm enjoys theoretical guarantees on smaller worst-case relative errors.

3, Experiments on various applications are good to support their findings.

Cons:

1, Personally, this paper is not easy to follow even though I’m quite familiar with random features. For example, I understand the first part in Sec. 2.3, e.g., Eq. (14), (15) till reading Sec. 2.3.1; The notation SM^{hyb}_{m,n}(x,y) is not defined until Lemma 2.4; Some notations are unclear. I suggest the author carefully polish this paper.

2, the relationship between Sec 2.2 and the algorithm in Sec. 2.3 is not clear to me. The authors attempt to unify trigonometric and positive random features under a same framework, i.e., SM^{cexp}. However, the proposed algorithm in Sec. 2.3 to approximate softmax kernel is the exact combination of them. The complex exponential estimator appears unused for Gaussian/softmax kernel approximation. More importantly, this estimator is asymmetric, which appears strange. This only holds for two special cases (symmetric due to A=(A^T)^{-1}) by A:= iI_d or I_d. This estimator appears invalid/asymmetric for general case. In fact, the asymmetric property destroyed the foundation of kernel associated with RKHS. In this case, this estimator (the approximated kernel) under a general matrix A is not well-defined in kernel methods.

3, In theory, Theorem 3.1/3.5 is not clear to show that whether SM^{anghyb} achieves variance reduction or smaller MSE when compared to SM^{trig} and SM^{++}. Refer to a survey with discussion on the variance reduction:

[S1] Liu, F., Huang, X., Chen, Y., & Suykens, J. A.K. (2020). Random Features for Kernel Approximation: A Survey on Algorithms, Theory, and Beyond. TPAMI 2021.

4, it’s good to see many experiments but evaluation on kernel approximation error is needed. Experimental results in Figure 3 is not enough. In fact, kernel approximation error (as well as unbiasedness) is quite important in random features based algorithms. To validate the reduction on approximation error, it would be better to compare the proposed algorithms with SM^{trig} and SM^{++} as well as other representative algorithms including ORF and QMC on typical datasets.
(ORF and QMC can be also used for softmax/Gaussian kernel approximation. The proposed algorithm might be inferior to ORF and QMC due to the standard sampling strategies. It's ok for me but at least it can beat SM^{trig} and SM^{++} on typical datasets.)

Besides, regarding to the experiments on Transformers for language modelling, the following refs is missing.

[S2] Peng, H., Pappas, N., Yogatama, D., Schwartz, R., Smith, N., & Kong, L. Random Feature Attention. ICLR2021.


**Summary Of The Paper:**

Author rebuttal checked. I increase my score to 8. BTW, I strongly suggest the authors polish this paper as the current version appears uneasy to follow, especifically for researchers unfamiliar with RFF.
=============================================================
This paper proposes a hybrid random features that combines the classical trigonometric and positive random features, which aims to achieve good approximation performance for softmax kernel with both small and large values. This framework is also applicable to Gaussian kernels. Numerical experiments on softmax kernel approximation in linear attention and robotics are conducted to support their algorithm and theory.

**Summary Of The Review:**

In sum, this paper provides a novel random features algorithm with theoretical guarantees and has good applications. Nevertheless, some parts on algorithm and experiments are unclear, which would devalue this paper. I would like to see the response from the authors

---

> ### Author Response · Authors · 2021-11-16
> **General comment**
>
> We would like to sincerely thank the Reviewer for the review and insightful feedback. We have uploaded the new version of the paper that addresses Reviewer’s comments.

---

> ### Author Response · Authors · 2021-11-16
> **Notation**
>
> Thank you very much for the comment. We have improved the notation in the most recent version of the manuscript. In particular, we defined $SM^{hyb}_{m,n}(x,y)$ before Lemma 2.3 (old Lemma 2.4) as the Reviewer suggested.
>
> Relationship between old Sec 2.2 (now Sec. 2.3) and the algorithm in old Sec. 2.3 (now Sec. 2.2):
>
> Old Section 2.2 (now Section. 2.3) provides a general recipe for building base estimators (beyond the trigonometric one and the one applying positive random features) for the hybrid constructions. Thus old Section 2.3 (now Sec. 2.2), that introduces HRFs, leverages constructions from the old Section 2.2 (now Section 2.3). It is true that the Gaussian and angular HRF variants still rely on the trigonometric and positive random feature estimators as base ones. However the other class of HRFs leveraging clustering information (that was also tested in the experimental section and provided gains over baselines) directly utilizes other base estimators introduced in old Section 2.2 (now Section 2.3).

---

> ### Author Response · Authors · 2021-11-16
> **Asymmetry of the general complex estimator**
>
> Thank you very much for the excellent question. We would like to clarify that the asymmetry is not a problem (as we also clearly see in the experiments involving clustering-based HRFs for Speech data and Gaussian clusters). The approximate kernel matrix still enjoys low-rank decomposition (which is a gateway to all the computational gains) even though it is not symmetric with probability 1 but only on expectation. Besides, for each entry of the approximate kernel matrix we can provide concentration results, leading to the concentration results for the entire approximate kernel matrix. We want to emphasize that the concentration results regarding kernel approximation (that lead to several downstream guarantees) *do not require* the approximating function to be symmetric with probability one, even though it approximates a symmetric function. Indeed, most results in the literature on the guarantees of the random feature maps (also for downstream applications such as ridge regression) do not leverage the symmetry since it is not needed (see for instance: “Random Fourier Features for Kernel Ridge Regression: Approximation Bounds and Statistical Guarantees”). The approximate function is stochastic and thus in general *does not even need to be globally well-defined* (across optimization steps), even for baseline random feature map mechanisms. Take for instance the Performer architecture with redrawing turned on (random features are resampled at each iteration). In this case the function takes different values for the same inputs for different optimization steps (even though the consistency is maintained within a fixed step), so in practice we have a series of different kernel functions used in different optimization steps. That clearly violates the standard kernel interpretation of the attention mechanism, but the method is successfully applied since it does not rely on this interpretation. All the desired features of the function (being well-defined, symmetric) are of course achieved on expectation, but the same can be said about the general complex estimator.
>
> Asymmetry would only be a problem (only from the practical point of view) if the particular application of the method would require perfect symmetry in treating both inputs to the kernel function, so from the context it would not be clear which random feature map mechanism to apply for some of the inputs. In all the interesting applications of the general complex estimator that is however not the case. Even more, in the important class of applications for attention-based methods, the symmetry between the inputs is broken by definition (even before any random feature map mechanism is applied) since the first input to the kernel function corresponds to query and the second ones to key (note also that because of the above, for all attention-based methods, the attention matrix in general *is not symmetric*). For multi-class classification, this is also the case and we clearly have two types of inputs: corresponding to embeddings of the classes and embeddings of the objects to be classified.

---

> > ### Comment · Reviewer_Ks6B · 2021-11-26
> > **increase the score to 8 but carefully polish the paper on asymmetry**
> >
> > Thanks for your update.
> > I have increased my score to 8, but I strongly suggest the authors polish this paper as the current version appears uneasy to follow, especifically for researchers unfamiliar with RFF.
> >
> > Besides, regarding to asymmetry, the reponse appears still unsatisfactory to me as it still cannot address my concern. I understand that, albeit asymmetry, the approximated matrix still enjouys low-rank decomposition. However, what's function space associated with an asymmetry kernel (matrix)? This is not associated with a finite-dimensional RKHS, where the finite-dimension is the number of random features.

---

> > > ### Author Response · Authors · 2021-11-27
> > > **Additional clarification**
> > >
> > > Thank you very much for the comment. We will further polish the paper to make sure that in the camera-ready version the RFF part is easy to follow for readers that are not familiar with random features.
> > >
> > > Regarding asymmetry, we focus in the paper on estimating softmax/Gaussian kernel values via the linearization trick (that gives rise to low-rank decomposition and consequently, space and time complexity gains) to obtain small mean squared error of the pointwise estimation. This in principle does not require to approximate a symmetric function with a symmetric one and furthermore we never claim that the most general estimator from our class needs to be interpreted as a *symmetric* kernel. This is not needed for theory to work (in particular, as we show, the asymmetric estimator can be chosen to have vanishing or close to zero variance of the estimation in the desired regions of space and asymmetry is a key to make it work) and not needed in practical applications, as we showed in the experimental section.
> > > Even though in general asymmetric, all considered functions are *on expectation* perfectly symmetric.
> > > We want to emphasize though that the asymmetric estimators can be interpreted as *asymmetric random kernels* (approximating the symmetric ones and providing symmetry on expectation). The theory of the asymmetric kernels in machine learning is well-developed, see for instance:
> > >
> > > 1. Asymmetric Kernel Learning; Wu et al.
> > > 2. Symmetric and antisymmetric kernels for machine learning problems in quantum physics and chemistry; Klus et al.
> > > 3. Support Vector Classifier with Asymmetric Kernel Functions; Tsuda
> > >
> > > In particular, the asymmetric kernels were successfully adapted to several ml algorithms were positive definite kernels were so far used on a regular basis, often outperforming them (see for instance: SVM algorithm in [3]).
> > > Finally, let us emphasize that not all hybrid random features estimators are asymmetric. A large impactful class represented in particular by the angular hybrid estimator is a class of symmetric functions.
> > >
> > > We completely agree with the Reviewer that the manuscript would benefit from further discussion on the asymmetry and thus will add this detailed explanation to the camera-ready version of the paper.

---

> ### Author Response · Authors · 2021-11-16
> **Theorem 3.1-3.5 and variance reduction guarantees**
>
> Thank you very much for the comment. Let us clarify that Theorem 3.5 says that the maximal relative error (defined as the ratio of the square root of the MSE to the approximated value) for the angular hybrid mechanism is strictly smaller than the one for baselines under similar computational budgets. We explain in detail in the paper why this metric is crucial to evaluate different estimators (see for instance: the paragraph before Lemma 3.4). The MSEs themselves are not the best metrics for several reasons. First of all, as explained in the paper: “Rethinking Attention with Performers”, the MSE of the FAVOR+ mechanism is not always smaller than the MSE of the trigonometric mechanism, yet the former is much better than the latter. Finally, it is not the absolute but relative error that is crucial. An estimator might be characterized by a small MSE but in the regions of space where it is estimating values orders of magnitude smaller than the standard deviation of the estimation. In this case, a small MSE is meaningless.

---

> ### Author Response · Authors · 2021-11-16
> **Experiments**
>
> Let us emphasize that the ORF mechanism is another technique for variance reduction that can be applied to *all the methods discussed in our paper*, also the HRFs. Thus for apple-to-apple comparison, one would need to either turn on ORF in all the compared algorithms or to turn it off. We have decided to turn it off since the paper is not about the way how the ensembles of random projections are chosen (which is what ORF is about). We also explicitly comment on that in the Related Work section (see for instance: “rather than focusing on improving sampling mechanism for a given approximation algorithm..”) Standard QMC methods (e.g. Halton sequences, etc.) is a class of techniques different from those applying random features (in particular they do not provide unbiased estimation) and thus are not a subject of this paper. Furthermore, as for ORFs, they focus on the way the random samples are constructed, which is not the subject of this paper.
>
> Having said that, following Reviewer’s request, in the updated version of the paper, we have compared QMC methods based on random Halton sequences with different random feature map mechanisms (HRFs and orthogonal random features), clearly demonstrating the advantages of HRFs (see: paragraph “​​Comparison with QMC-methods:” in Section 4.1).

---

> ### Author Response · Authors · 2021-11-16
> **Additional references**
>
> In the updated version of the paper we have included references to: “Random Features for Kernel Approximation: A Survey on Algorithms, Theory, and Beyond” and “Random Feature Attention”.

---

> ### Author Response · Authors · 2021-11-27
> **General Comment**
>
> Thank you very much for reading the rebuttal and additional suggestions. We will further clarify the exposition in the camera-ready version to make it easier to understand by a person not familiar with random features. We will also add additional discussion regarding asymmetry (see also one of our new comments below).

---

### Official Review · Reviewer_iboB · 2021-11-01

**Correctness:** 4
**Technical Novelty And Significance:** 4
**Empirical Novelty And Significance:** 2
**Recommendation:** 8
**Confidence:** 4

**Main Review:**

**The contributions of the paper are original and very well motivated**, as they aim at providing a scalable and unbiased estimator of the widely used softmax and gaussian kernels which also has low variance.
Namely, the principal contributions are:
1. Hybrid Random Features as composition of base estimators, unbiased, that adapt to data points and can be linearized;
2. Generic form of base estimators, named complex exponential estimator;
3. Three instantiations of HRF that provide low variance estimators.

The proposed HRF can benefit the ICLR community at large and in particular the fervent research around transformers.
The theoretical results seem sound, however I haven't checked in details the proofs of Lemma 2.4, Lemma 3.2, Lemma 3.4 and Theorem 3.4.

**The paper itself is hard to read** in particular because intuitions and justifications for the derivations and definitions are rarely provided.
I have few suggestions for improving its presentation:

1. The ordering of the exposition of the contributions does not help the reader understand the train-of-thoughts and intuitions. I'd suggest to first introduce HRF (Section 2.3) to then show how base estimators can be chosen (Section 2.2) and how $\lambda$-coefficients can be defined accordingly.
Moreover, it would help the presentation of the complex exponential estimators to start from defining the new feature map $\Phi_M^m(u)$'s components $\exp(wMu)$ to clarify first the goal of defining a generalized form of random features that can be specialized to well known ones depending on $M$. In the current presentation the new choice of $z$ comes out of the blue.

2. When presenting HRF, the choice of having kernels as $\lambda$-coefficients and not e.g. constants is not motivated and not sufficiently highlighted. I understood only later, when examples of instantiations are provided, the interest of this choice which is to reduce the variance of the estimator. Still it is not clear to me why they need to be shifted (with the offset parameter $a_k$) and whether in practice this choice provides better results than fixing these coefficients to constants, e.g. learned from a subset of data.

3. Maths and notation could be simplified. Namely, Section 2.3 could be written for $p=1$ (2 base estimators), and the general form for $p > 1$ could be reported in the appendix.

4. When presenting the theoretical results, it should be made more explicit which results are new and which are not. This is not clear especially for Lemma 2.3. A reference for Definition 2.1 should be provided and in general pointers to the proofs in the appendix should be added.

5. The experimental section of the main text is too packed. It is valuable to have carried out the empirical analysis on multiple tasks and contexts, but reporting them all comes at the cost of losing clarity, as experiments and results are not thoroughly described and commented.
More precisely:
(a) Figure 1: colorbar is missing and it is not clear why the light blue is both for low and high MSE values.
(b) Section 4.1: What study is referred to as complete ablations? Why the results for gaussian $\lambda$-coefficients are not reported? What is the number of random features for Angular Hybrid?
(c) Section 4.2: Why is $s$ a vector of size 4 when $l(A)$ should be a scalar?
(d) Section 4.3: Which task is performed here and what is it meant by negative classes? Are the differences between methods for the three metrics significant? What is the perplexity score? A note should be added on why negative fractions are not reported for FAVOR+.
(e) Section 4.5: Why is training with state-of-the-art estimators unstable on this task (loss going to Nan for particular numbers of random features)? What are the "drastic compute improvements" given that score differences between methods are not significant for the same number of iterations?
I'd suggest to select the most important experiments and present them thoroughly in the main text, while reporting the rest in the appendix.

Minor:
1. MSE in this context should be defined in the introduction.
2. Gaussian kernel's inputs are missing in Eq. (1).

**Summary Of The Paper:**

The paper proposes a new type of estimators of softmax and gaussian kernels based on random features, consisting of compositions of base estimators such as the trigonometric and positive random features. The composition is a linear function, such that the new estimator (called hybrid) is unbiased, whose coefficients (called $\lambda$-coefficients) are independent of the base estimators and are also kernel functions (to better adapt the estimator to the compared data) hence also need to be linearized for scalability.

Three different instantiations of the hybrid estimator are proposed using base estimators of the form of the defined complex exponential one (an asymmetric generalization of random feature estimators). In particular the angular one (i.e. $\lambda$-coefficients depending on the angle between the pair of data points) is shown to have low variance and low maximal relative error for small and large kernel values.
HRF has also a lower computational complexity in the total number of random features w.r.t. non compositional estimators.

Experiments are carried out on multiple applications, first to verify the approximation capability of the hybrid estimator w.r.t true kernel, and trigonometric and positive estimators in simple settings, then to show the improvement (either in quality scores or computational) of using the proposed estimator in models requiring the use of softmax at scale.

**Summary Of The Review:**

The paper makes strong contributions with potential high impact, however its presentation is over-complicated and lacks clarity. I suggest to improve on this point and I am willing to increase my score accordingly.

## UPDATE AFTER REBUTTAL
The presentation of the paper has improved and, while some notation are still hard to read, I believe that the paper's contributions are now more accessible. I am hence increasing my score to 8. I am looking forward to seeing how the paper will be presented in video lectures.

---

> ### Author Response · Authors · 2021-11-16
> **General comment**
>
> We would like to sincerely thank the Reviewer for the review and insightful feedback. We have uploaded the new version of the paper that addresses Reviewer’s comments.

---

> ### Author Response · Authors · 2021-11-16
> **Exposition of the paper**
>
> Thank you very much for the comment. We have made the suggested changes in the new version of the manuscript in order to improve the presentation, in particular:
>
> (1) We have changed the order of the old sections: 2.2 and 2.3 to first introduce HRF and then explain how base estimators and the corresponding lambda-coefficients can be chosen.
>
> (2) While introducing complex estimators, we start now by defining the new feature map components to clarify the goal of the generalized form of random features with special instantiations corresponding to well-known random feature map mechanisms given later, as the Reviewer suggested (see: comment above Equation 17 in the updated manuscript).
>
> (3) We clarified why lambda-coefficients are defined as (potentially shifted) kernels. This choice enables us to linearize the entire estimator as long as those kernels can be linearized (as we explained at the beginning of the paper), and thus is a gateway to all the computational gains. We in fact pointed this out already in the original version of the manuscript (see: discussion above original Lemma 2.4 (now Lemma 2.3)), but do it even more explicitly in the current manuscript (see:  the comment under the formula for the lambda-coefficients: “Linearization of the $\lambda$-coefficients given by Equation 9 is crucial to obtain linearization of the hybrid estimators, and consequently random feature map decomposition.”). Choosing lambda-coefficients as arbitrary or trainable functions of keys/queries in general does not provide linearization and making them some absolute (potentially trainable constants) is not an expressive enough mechanism to reduce the variance of the estimation. The coefficients should be chosen in such a way that different base estimators play more important roles in the overall hybrid construction in different regions of space determined by the landscape of their corresponding variance functions which depend on queries/keys. The shifted variant is general enough to cover the angular mechanism which elegantly leverages the information about the angle between the query and the key for variance reduction. Without this additional affine term, it would not be possible.
>
> (4) Thank you for the comment on rewriting old Section 2.3 (now Section 2.2) to consider only the case p=1 and included the most general mechanism in the Appendix. We have considered this, but have decided  not to do it, since we have included additional important experiments leveraging the full strength of the general HRF mechanism in the most updated version of the manuscript. Please see our general comment to all the reviewers for more details (“General comment to all the reviewers”). However we have improved this section, in particular by adding additional intuitions (e.g. the role of lambda-coefficients, see: “The role of the $\lambda$-coefficients is to dynamically (based on the input $(\mathbf{x},\mathbf{y})$) prioritize or deprioritize certain estimators to promote those which are characterized by lower variance for a given input.”) as well as by adding an additional paragraph on the important special case of p=1 (bipolar estimators).
>
> (5) Theoretical results & proofs: Whenever we did not say explicitly that the result is taken from the other paper, the result is new. That means that the only theoretical result that is not new is Lemma 2.2 as was always clearly stated in the paper (see the comment right before Lemma 2.2: “The following result from (Choromanski et al., 2021b) shows that the mean squared error (MSE) of the trigonometric estimator is small for large softmax kernel values and large for small softmax kernel values, whereas an estimator applying positive random features behaves in the opposite way”). We further clarified in the current version of the manuscript (see: beginning of Section 2).
>
> (6) We de-densified the experimental section to report the results on the subset of tasks (with more detailed description; see for instance description in new Section 4.3) while including other results in the Appendix (e.g. we moved the entire section on applying clustering-based HRFs on the synthetic datasets as well as former Fig. 4 to the Appendix).

---

> ### Author Response · Authors · 2021-11-16
> **Additional clarifications**
>
> (1) Fig. 1 and colorbars: Thank you very much for the  comment. To construct this figure, we used a standard online library for creating good-quality 3d-plots of 2d mathematical functions (since we have explicit formulae for the MSE): https://www.math3d.org. The color-code applied there was something we did not have control over.
>
> (2) By complete ablations in Sec 4.1 we mean ablations over even more lengths of queries/keys (see: figure in Sec. G of the Appendix with additional lengths). We clarify it in the most updated version of the manuscript. The reason why we moved r=1.25 to the appendix and only kept r=1 and r=1.5 in the main text is due to the space limit.
>
> (3) The results for the variant using Gaussian coefficients were similar to those for the angular (in particular in both settings we have gains over baseline estimators). We clearly state it in the current version of the manuscript (see: description of Fig. 2 in the new version of the manuscript). In the paper we show different ways of constructing hybrid estimators, yet mentioning at the same time that some are better than others. Gaussian λ-coefficients could also achieve smaller relative error on  close to 0 and  compared with Trigonometric and FAVOR+, even though not as good as the angular hybrid estimator. We clearly explained it in the paper (see: “The mechanism can be thus trivially applied to the setting of accurate approximation of both small and large softmax kernel values for inputs of fixed length, yet it turns out to be characterized by the larger variance than its angular hybrid counterpart and cannot provide variance vanishing property for both: theta(x,y) = 0 and theta(x,y) = π.”). Given all the above (in particular similar results for both the estimators as compared with the baselines) and the fact that the experimental section is already very dense, as the Reviewer pointed out, we have decided to include only the angular version.
>
> (4) The number of random features is chosen to provide a similar number of FLOPS operations for fair comparison, as clearly stated in ALL corresponding experiments (see: figures with the descriptions).
>
> (5) The size of the vector s: Thank you for your question. Vector s is of size 4 since we have: two clusters for inputs x (queries), two clusters for inputs y (keys) and furthermore:  for every pair (query-cluster, keey-cluster) we have a customized random feature map mechanism.
>
> (6) In Section 4.2 (old Section 4.3) we perform a language modeling (LM) task described in “Sampled Softmax with Random Fourier Features” (see: Section 4.1 and 4.2 in that paper), i.e. training a language model using LSTM, where the normalized output of the LSTM serves as the input embedding.
>
> (7) The term “negative classes” refers to all the classes other than the true class. This definition is used in the paper we cited in this section (“Sampled Softmax with Random Fourier Features”). We clarify it in the current version of the manuscript.
>
> (8) The perplexity score (a common metric to measure the training quality of language model) is defined as 2^{cross entropy loss}. We also clarify it in the updated version of our manuscript (Appendix Section I).
>
> (9) Negative fractions are not reported for FAVOR+ since they are zero by the definition of positive random features (the dot-product of two vectors of positive entries cannot produce negative value). We clarified it in the current version of the manuscript in the new description of Fig. 3 (old Fig. 4).
>
> (10) in Section 4.4 (old Section 4.5) the unstable behaviour of the base estimators is caused by their large variance for the given number of random features used. Increasing this number reduces instability, but at the cost of the additional unacceptable computational time. We clarified it in the current version of the manuscript in the description to Fig. 5 (old Fig. 6).
>
> (11) The *drastic computational improvements* come from the fact that processing each iteration takes much less time for the hybrid estimator (since it uses many fewer random features). Thus even though the number of iterations is similar for a particular level of performance, the absolute wall-clock time is much smaller. We already explained it in the first version of the manuscript, where we reported 3x fewer FLOPS for the HRF method in the description of Fig. 5 (old Fig. 6). We clarified it in the current version of the manuscript (see: Section 4.4).

---

> ### Author Response · Authors · 2021-11-16
> **Minor comments**
>
> In the current version of the manuscript we clarified that MSE (mean squared error) is just the variance of the estimator since all considered estimators are unbiased, see: “We denote by MSE the mean squared error of the estimator (i.e. its variance since all estimators considered in this paper are unbiased)” in the Introduction section (in the paragraph after Eq. 2 and before Eq.3, before this notation is applied first time in the paper).
> We also put the Gaussian kernel’s inputs in Eq. (1), thank you for catching this typo !

---

> ### Author Response · Authors · 2021-11-28
> **Feedback on the rebuttal and the updated version of the manuscript**
>
> We would like to sincerely thank the Reviewer for the feedback and very insightful comments. We did our best to address *all* the comments and updated the manuscript accordingly. Any feedback on the rebuttal and the new version of the manuscript before the discussion period ends will be greatly appreciated. We want in particular to make sure that we can promptly address potential additional Reviewer's comments before discussion period ends.

---

> > ### Comment · Reviewer_iboB · 2021-11-29
> > **update**
> >
> > I thank the authors for addressing my remarks and improving the presentation of the paper. Please find my review updated above.

---

> > > ### Author Response · Authors · 2021-11-29
> > > **Thank you for your feedback**
> > >
> > > We once more thank the Reviewer for the very valuable and detailed feedback that helped us to improve the presentation of the paper.

---

### Official Review · Reviewer_YmNP · 2021-11-05

**Correctness:** 3
**Technical Novelty And Significance:** 3
**Empirical Novelty And Significance:** 3
**Recommendation:** 6
**Confidence:** 4

**Main Review:**

The main idea of the paper – to try to get the “best of both worlds” when different types of random features are useful in different regimes – is very nice. The theoretical results show that HRFs can be superior to either of the “base” random features used. The experiments in section 4.1 on pointwise kernel estimation also nicely support this narrative. The other experiments suggest at least some benefits, with perhaps the strongest results being for the speech model example in section 4.4. However, there are a few serious issues that need to be addressed.

## Writing and presentation

This was a very confusing paper to read.

### Notation

There is an incredible amount of very dense notation, much of which it should be possible to drastically simplify with a little thoughtful effort. For example, there are many cases where to approximate a function $f(x, y)$, a random feature representation of the form $\mathbb E[\psi^{(1)}(x)^\top\psi^{(2)}(y)]$ is used. But rather than, as I have done, reuse the same symbol for both and distinguish the random functions with an index, different symbols are used (e.g., $\alpha$ and $\beta$). This forces the author(s) to write about $\phi$, which can be either $\alpha$ or $\beta$. Thus, whereas a single symbol plus an index would do, the reader instead has to juggle three symbols. For a particularly egregious example of this, see the sentence that begins before eq. (15).

Another reason for the difficult notation is the author(s) try to make the methods presented as general as possible, which puts a heavy burden on the reader. However, often this generality is now actually used very much. For example:

* Nearly all of the focus is on “bipolar” HRFs with just $p + 1 = 2$ “base” random features. However, the general case is covered, which requires, e.g., to define $\lambda^k$ for $k=1,\dots,p$, whereas for the most part only a single $\lambda$ function is needed.
* $\lambda^k$ can take values in $\mathbb R$, but in all examples it is constrained to [0,1]


### Paper structure

The paper needs to be restructured on both local and global scales. At the local scale, definitions and semantic explanations often come long after something is used. For example, what is the purpose of $\lambda^k$? It is unclear until a few pages after it is defined. Or, the definitions of $\otimes$ and $\prod^\star$ in Lemma 2.4 are at the very end, leaving the reader unsure what any of the four displayed equations that come before actually mean. Sometimes objects are never defined (e.g., MSE and $\mathcal P(\mathbb R^d)$) or only long after they are first used (e.g., $\boldsymbol i$ is finally defined right before Lemma 2.4). Also, global definitions are put inside lemmas and theorems (e.g., Lemma 2.2).

At the global scale, the main contribution of trading off whether the trig or positive random features are “active” is in the paragraph at the top of p. 5. If some version of this came, say, after section 2.1, followed by the material in section 3 (theoretical guarantees), the reader would have everything they need to understand all but the experiments in section 4.2, which in any case aren’t very interesting.

### Use of colons

There is liberal use of colons in the middle of sentences. Please remove these.

## Scope of contribution

There are a number of places where I don’t think the paper properly describes the scope of its contribution.

1. Abstract: “By generalizing Bochner’s Theorem for softmax/Gaussian kernels and leveraging random features for compositional kernels, the HRF- mechanism provides strong theoretical guarantees”. Given the generality of Bochner’s theorem, this sounds like a major contribution and an important part of what makes HRFs work. However, I believe the generalization of Bochner’s theorem is Lemma 2.3, which seems to only be used for the HRFs for clustering structure; these make only a brief appearance in the paper: there is no theory for them and they earn a small synthetic experiment (Section 4.2), but there practical utility is never shown. Thus, Lemma 2.3 appears to have nothing to do with the “strong theoretical guarantees” developed in section 3, which are only for the bipolar kernel with trig and positive random features.

2. Related work at the end of section 1: “rather than focusing on improving [the] sampling mechanism for a given approximation algorithm, [HRFs] provide a completely new algorithm.” I find this claim to be misleading. HRFs provide a way to combine existing random feature algorithms.

3. Generality. As I’ve mentioned above, the paper presents HRFs in great generality, but very little of that generality is actually used in a consequential way. It may be that this generality is quite useful! I really like the idea, and could imagine othe HRFs will be developed or the cluster-based HRF will find use cases. But the theory and experiments given in the paper don’t support such grand claims about “strong theoretical guarantees” for the “HRF mechanism.” On the contrary, the paper shows that one particular instantiation of the HRF mechanism has smaller relative MSE than the underlying components.

## Experiments

Section 4.1: This was a nice clean demonstration of the theory. I think the red dotted lines in Fig. 2 are for zero. Please clarify this in the caption.

Section 4.2: As mentioned above, unless this experiment is supported by some practical application, it should be moved to the appendix as a “proof-of-concept”.

Section 4.3: It seems like none of the random feature approaches give very good approximations (Fig. 4). There are no error bars in Fig. 5, so I’m unsure if the differences are meaningful. Moreover, I’m not convinced the ordering would remain the same if the experiment were run for, say, 10 more epochs.

Section 4.4: This is a potentially strong experiment. However, the statement that “ ≥ 0.2% are considered to be statistically significant for that task” makes no sense. The differences are either statistically significant or not, given a particular test and choice of test size.

Section 4.5: The results here are promising. A 3x improvement in FLOPS certainly is something. But the lower variation in reward, particular in the step-stone locomotion task, is also noteworthy.

## Recommendations

1. Simplify the notation

2. Focus only on bipolar random features in the main text and move the other types and the general definitions  (e.g., material around eq. (14), Gaussian lambda coefficients, adaptation data admitting clustering structure, section 4.2) to the SM or to a separate section that can be skipped on first reading.

3. Restructure the paper so motivation and semantic interpretations come before definitions

4. Clarify scope and contributions.

5. Strengthen experiments by including error bars for all performance metrics and running the experiments in section 4.3 for at least 10 more epochs, or until the perplexities stabilize.


**Summary Of The Paper:**

The paper makes two methodological contributions: a new approach to constructing randomized approximations to Gaussian and softmax kernels and a proposal to combine multiple randomized approximations to create “hybrid random features” (HRFs). The idea of the hybrid random features is to activate a particular approximation for pairs (x,y) for which that approximation is accurate. Some theoretical results support the use of a particular instance of HRFs that combine trigonometric and positive random features, showing the hybrid has lower mean squared error than either type of random feature. Experiments are presented to verify the benefits of HRFs on a variety of tasks, with a focus on their use in neural network settings.

**Summary Of The Review:**

A potentially useful method with a nice idea and some promising results. But the paper structure and writing need to be improved, the scope and contributions clarified, and the experiments tightened up.

---

> ### Author Response · Authors · 2021-11-16
> **General comment**
>
> We thank the Reviewer for the comments. We have uploaded the new version of the paper that addresses them.

---

> ### Author Response · Authors · 2021-11-16
> **Notation**
>
> Thank you for the comments. We have made the suggested changes regarding random feature map notations in the new version of the paper. Note that the suggested changes lead to some text reduction, but at the cost of introducing additional indices. Since the paper requires a very technical notation for rigorous mathematical presentation (which we definitely do not want to give up), the notation cannot be simplified too much. Please, see also our general comment to all the reviewers (“General comment to all the reviewers”).
>
> In the updated version of the paper we clarified that lambda^k takes values in the [0,1]-interval.

---

> ### Author Response · Authors · 2021-11-16
> **Paper structure**
>
> Following Reviewer’s advice, we restructured the paper in both a local and global sense.
>
> (a) Local scale:
>
> We made sure that the motivation and semantic interpretations come before definitions. In particular:
> (1) We clearly explained the purpose of lambda^{k} right after introducing lambda-coefficients in the current version of the manuscript (see: “The role of the $\lambda$-coefficients is to dynamically (based on the input $(\mathbf{x},\mathbf{y})$) prioritize or deprioritize certain estimators to promote those which are characterized by lower variance for a given input.”).
> (2) In the current version of the manuscript, we put the definitions used in Lemma 2.3 (previously Lemma 2.4) right before Lemma 2.3.
> (3) Mathematical Notation:  Even when we use standard mathematical notation, e.g. MSE, we provide an explanation right away. As a matter of fact, in the original version of the paper we explained that MSE stands for the mean squared error the first time we used this shortcut. In the current version we furthermore emphasize that since all the estimators under consideration are unbiased, MSE equals to variance.  We assume that we do not need to further explain certain fundamental concepts in probability theory such as mean squared error or variance since reviewers who are confident about their evaluation of this paper clearly (by definition) need to be familiar with all basic concepts from probability theory for accurate evaluation of this manuscript. Furthermore, when P(R^{d}) is first introduced, from the context it is completely clear that it refers to the set of probabilistic distributions over R^{d}, even though we clearly state it now to leave no doubt. In the current version of the manuscript, we define the complex number i at the very beginning of the section where we introduce complex estimators (before it is used for the first time), even though we want to emphasize that we apply here standard notation from the complex analysis.
> (4) In the updated version of the manuscript, we moved global definitions outside of lemmas and theorems.
>
> (a) Global scale:
>
> The Reviewer has suggested that the version of the paragraph from the top of p.5 (about the angular kernel) might have been placed right after old Section 2.1 and then this material could be followed by the results in old Section 3 (theoretical guarantees). The idea was to focus on the angular hybrid version in the exposition. This comment was based on the fact that the results for the clustering-based HRF variant were conducted only on the synthetic datasets. In the most updated version of the manuscript we have added clustering-based HRF method to our Speech benchmark (see: Section 4.4, previously Section 4.3), showing that it outperforms *all* other methods (also angular hybrid). Thus important real data experiments are now conducted also for the clustering-based HRF variant (not only the angular one) and therefore the clustering variant plays an important role in the experimental section. Thus we have decided not to conduct suggested changes in order to be able to include both: the angular and clustering variant within one coherent theoretical framework. See also our general comment to all the  reviewers (“General comment to all the reviewers”) for more details.

---

> ### Author Response · Authors · 2021-11-16
> **Use of colons**
>
> We fixed it in the current version of the paper.

---

> ### Author Response · Authors · 2021-11-16
> **Scope of contribution - part I**
>
> 1. Abstract: We think that the Reviewer might have misunderstood something. The statement “By generalizing Bochner’s Theorem for softmax/Gaussian kernels and leveraging random features for compositional kernels, the HRF mechanism provides strong theoretical guarantees - unbiased approximation and strictly smaller worst-case relative errors than its counterparts” is completely correct. HRFs rely on two pillars: generalized complex estimator mechanism that gives in particular the first unified view on the base estimators of the angular variant (for which worst-case relative error guarantees were explicitly given in Theorem 3.5) and thus provides a natural framework to combine them, as well as compositional kernels leading to required linearization of the hybrid mechanism (and consequently, desired computational gains). That combination provides a way to zero out the variance in several desired points in space and consequently, lower down the worst-case relative error. The unbiasedness of HRFs critically relies on the generalized complex estimator theory and the way base estimators are combined via the compositional mechanism.
>
> It is not true that worst-case relative error gains are obtained only for the variant using two known before base estimators and it is easy to see that. A trivial example is the setting, where one of the base general complex estimators has variance zeroing out for the linear transformation close enough to identity and the other close enough to its negated version. Since the worst-case relative error is trivially a continuous function of the hyperparameters of the general complex estimator, by Theorem 3.5, such a variant still enjoys strictly better worst-case relative error. This variant applies as base estimators constructions that were not known before. We want to point out that this is not just a neat theoretical construction. In practice we do not necessarily want to zero out the variance for theta = 0 and theta = pi and the particular choice of base estimators might be guided for instance by data clustering process (for instance, small attention matrix entries will never correspond to theta=pi but to some theta close to pi and this more detailed knowledge might be leveraged during the process of assigning base estimators). In fact in old Section 4.2 (these results were now moved to the Appendix) and in new Section 4.3 we demonstrate that looking at data clustering properties is in general a right approach to achieve this goal. Obviously the related HRF-based estimators will rely on the most general version of the complex estimator. The result on the worst-case guarantees in Theorem 3.5 was presented only for the angular kernel for the clarity of analysis, but as explained above, this result trivially leads to similar guarantees for infinitely many other HRF variants. The corresponding detailed quantitative analysis would be the function of the particular hyperparameters of the HRF-estimator and their choice should ultimately be data-driven. Including such an analysis in the already very technical paper definitely would not improve the presentation.
>
> To summarize, the angular variant by no means is the optimal HRF version in terms of the worst-case relative error guarantees (that would question the need for the general HRF construction). It  is just one of the simplest (since its hyperparameters do not depend on data statistics) HRF variants providing strict theoretical gains over baselines.
>
> In the updated version of the manuscript (see: Section 4.3), we provided additional results on Speech data, showing the superiority of the general clustering-based HRF mechanism over the angular one, fully exercising the strength of the general HRF construction.
>
> 2. Related Work: “rather than focusing on improving [the] sampling mechanism for a given approximation algorithm, [HRFs] provide a completely new algorithm.”
>
> HRF is not just a novel way to combine existing random feature map mechanisms, but it also introduces novel base estimators to combine (general complex mechanism). This is demonstrated also empirically in experiments involving Gaussian clustering (these results are now moved to the Appendix) as well as new experiments for Speech data (Section 4.3), see the comment above. What we meant in that sentence is that we abstract in the paper from the way how random samples for Monte Carlo estimation are constructed (by definition they are chosen iid but one can also use orthogonal random projections as in ORF-methods or other related QMC methods). We clarify it in the most recent version of the manuscript.

---

> ### Author Response · Authors · 2021-11-16
> **Scope of contribution - part II**
>
> 3. Generality:
>
> We addressed this in detail in the first set of comments in the Scope of contribution section. We explain there in particular the importance of our theoretical results that cannot be reduced to a single concentration result for a particular hybrid variant. We also refer to additional experiments with Speech data showing the full strength of the general HRF construction and its advantages over constructions using known base estimators (such as the angular variant).

---

> ### Author Response · Authors · 2021-11-16
> **Experiments**
>
> 1. We have included additional experiments with Speech data showing the full strength of the general clustering-based HRF construction and its advantages over constructions using known base estimators (such as the angular variant). We have also added variance information for all tested methods.
> 2. In the language modeling section, we moved an updated figure on end-to-end training (old Fig. 4) (trained for extra 30 epochs, with std added averaged over 25 instead of 10 runs as before) to the Appendix since after additional training different methods perform similarly. We still want to point out that HRFs have better stats regarding softmax kernel estimation quality (e.g. Wasserstein distance, see: Fig. 3). Note that end-to-end training in principle does not need to fully leverage this, since other non-softmax sampling distributions may also in principle lead to good quality models.
> 3. We confirm that the red dotted lines in Fig. 2 are for zero and clarify this in the caption in the most recent version of the manuscript.
> 4. Old Section 4.2 (now in the Appendix): as discussed above, we have added additional results regarding the general clustering-based HRF mechanism for the Speech task in Section 4.3 (showing that it outperforms the angular hybrid variant and thus is also better than all other variants). It is currently in the main part of the paper and the results on using clustering-based HRFs for the synthetic datasets were moved to the Appendix.
> 5. Old Section 4.3 (now Section 4.2): In all the subfigures lower numbers are better. In the first subfigure of Fig. 3 (old Fig. 4) the hybrid and trigonometric mechanisms provide much better results than FAVOR+ and hybrid is consistently beating trigonometric in all the metrics.
> To understand the possible range of Wasserstein distance values, note that it is given as:
> $W_{d}(\mu, \nu ) \leq diam(A)TV(\mu, \nu )$, where A is the support of the measures $\mu$ and $\nu$, and $TV(\mu, \nu )$ is the total variation of the measures. Without any additional assumptions, $TV(\mu, \nu )$ can be bounded by $1$ and diam(A), where the latter in the case for the Penn Tree Bank dataset is $10000$ (the size of the vocabulary), leading to the upper bound 10K.   Thus the quality of the estimation of the original softmax distribution (Wasserstein distance < 1500) provided by the hybrid mechanism is high.
>  In terms of the negative fraction subfigure, the angular variant provides substantially better results than the other estimators (15% negative fraction), in particular the trigonometric one. Given that the latter estimator is applied in that setting on a regular basis (see for instance: “Sampled Softmax with Random Fourier Features”), this is a strong result.
> In the language modeling section, we moved updated old Fig.4 (trained for extra 30 epochs, with std added averaged over 25 instead of 10 runs as before) to the Appendix since after additional training different methods perform similarly. We still want to point out that HRFs have better stats regarding softmax kernel estimation quality (e.g. Wasserstein distance, see: Fig. 3). Following the reviewer's request on error bar, we have rerun the experiment in Sec. 4.2 (previously Sec. 4.3) to make sure the results are statistically stable and added a table in the appendix to show the means and the standard deviations of all methods in different metrics. Note that end-to-end training in principle does not need to fully leverage this, since other non-softmax sampling distributions may also in principle lead to good quality models.
> 6. Old Section 4.4 (now Section 4.3): We want to clarify that what we meant here is that the difference between the hybrid estimator and the other ones is much larger than the standard deviation as measured across different training runs. We added this additional information regarding standard deviation in the most recent version of the manuscript (see: new Section 4.3).
> 7. Old Section 4.5 (now Section 4.4): Thank you for the comment. The variance of the reward is very similar for both the methods, but the speed improvements coming from the 3x reduction of FLOPS for the hybrid variant makes it a clear winner for these tasks.

---

> ### Author Response · Authors · 2021-11-28
> **Feedback on the rebuttal and the updated version of the manuscript**
>
> The authors would like to once more thank the Reviewer for the review. We did our best to address in detail *all* the comments and updated the manuscript accordingly several days ago. We do believe the new manuscript addresses Reviewer's concerns (see in particular section 4.3 where we present additional experiments on the clustering-based HRF mechanism for real data clearly demonstrating its advantages over other estimators ). We would sincerely appreciate any feedback on the updated version of the manuscript before the discussion period ends.

---

> > ### Comment · Reviewer_YmNP · 2021-11-29
> > **thanks**
> >
> > Thanks for your thoughtful responses and significant revisions to the paper. I think the new paper structure substantially improves readability. The new section 4.3 experiments provide sufficient evidence for the benefits of the clustering approach to justify the general treatment of HRFs in the main paper.
> >
> > My remaining concern is the claimed scope of the contribution. You should clarify in the abstract that the theoretical guarantees are for a particular instantiation of the HRF mechanism, not for HRFs in general. I also think it would be good to tease apart of the HRF mechanism itself and the possible ways to instantiate the base random features. The two are related (e.g., the clustering approach), but should still be distinguished in the abstract and introduction.

---

> > > ### Author Response · Authors · 2021-11-29
> > > **Thank you for your feedback**
> > >
> > > We would like to sincerely thank the Reviewer for the feedback. In the camera-ready version we will add suggested clarifications in the abstract. We will also distinguish in the abstract and the introduction the HRF mechanism itself from different ways of instantiating base estimators.

---

### Author Response · Authors · 2021-11-16
**General comment to all the reviewers**

In the updated version of the manuscript we have included additional important Transformer-training results on the Speech data (see: Section 4.3, previously Section 4.4) showing that clustering-based HRF estimator outperforms all other methods, also applying angular-hybrid estimators (that already outperform previous SOTA algorithms). These experimental results further validate our choice to present the HRF mechanism in its full generality, since it is now applied in its full generality in the experimental section on *real data*.
Since the clustering-based HRF method in the experimental section is not bipolar (we have: p>1) and does not apply an angular kernel, we cannot focus only on the angular hybrid variant and the bipolar setting (p=1) in the main body of the paper. Having said that, we have added a special paragraph on the bipolar estimator setting (in Section 2.2 in the updated manuscript) since it is a prominent special instantiation of the most general case.
We have conducted all other suggested changes regarding presentation of the results.

---

### Decision · Program_Chairs · 2022-01-20

**Decision:**

Accept (Poster)

**Comment:**

Kernel methods are among the most flexible and powerful approaches of our times. Random features (RF) provide a recent mechanism to also make them scalable due to the associated finite (and often small)-dimensional approximate feature map (in the paper referred to as linearization). The focus of the submission is the linearization of the softmax kernel (defined in (1)) while making sure that the obtained RF approximation is accurate simultaneously for the small and the large kernel values. The authors present a hybrid random feature (HRF, defined in (8)) construction parameterized by base estimators and weights, and show that specific choice of these parameters is capable of implementing the goal. Some of the HRF estimators are also accompanied by theoretical guarantees (Section 3). Their numerical efficiency is illustrated (Section 4) on synthetic examples and in the context of natural language and speech modelling, and in robotics.

Scaling up kernel methods is a fundamental task of machine learning. The authors present a nice and valuable construction in this direction which can be of both theoretical and practical interest to the community.

The submission would benefit from implementing the remarks of the reviewers to improve its clarity.